# Impact of high-resolution *a priori* profiles on satellite-based formaldehyde retrievals

Si-Wan Kim[1,2,3], Vijay Natraj[4], Seoyoung Lee[3], Hyeong-Ahn Kwon[5], Rokjin Park[5], Joost de Gouw[1,2], Gregory Frost[1], Jhoon Kim[3], Jochen Stutz[6], Michael Trainer[1], Catalina Tsai[6], and Carsten Warneke[1,2]

[1] NOAA Earth System Research Laboratory, Chemical Sciences Division, Boulder, CO 80305,
USA

[2] Cooperative Institute for Research in Environmental Sciences, University of Colorado, Boulder, CO, USA

[3] Department of Atmospheric Sciences, Yonsei University, Seoul, South Korea

[4] Jet Propulsion Laboratory, California Institute of Technology, Pasadena, CA, USA

[5] Department of Earth and Environmental Sciences, Seoul National University, Seoul, South Korea

[6] Department of Atmospheric Sciences, University of California Los Angeles, CA, USA

Correspondence to: Si-Wan Kim (siwan.kim@noaa.gov)

**Abstract**

Formaldehyde (HCHO) is either directly emitted from sources or produced during the oxidation of volatile organic compounds in the troposphere. It is possible to infer atmospheric HCHO concentrations using space-based observations, which may be useful for studying emissions and tropospheric chemistry at urban to global scales depending on the quality of the retrievals. In the near future, an unprecedented volume of satellite-based HCHO measurement data will be available from both geostationary and polar-orbiting platforms. Therefore, it is essential to develop retrieval methods appropriate for the next-generation satellites that measure at higher spatial and temporal resolution than the current ones. In this study, we examine the importance of fine spatial and temporal resolution *a priori* profile information on the retrieval by conducting approximately 45,000 radiative transfer model calculations in the Los Angeles Basin megacity. Our analyses suggest that an air mass factor (AMF, a factor converting observed slant columns to vertical columns) based on fine spatial and temporal resolution *a priori* profiles can better capture the spatial distributions of the enhanced HCHO plumes in an urban area than the nearly constant AMFs used for current operational products by increasing the columns by ~50% in the domain-average and up to 100% at a finer scale. For this urban area, the AMF values are inversely proportional to the magnitude of the HCHO mixing ratios in the boundary layer. Using our optimized model HCHO results in the Los Angeles Basin that mimic the HCHO retrievals from future geostationary satellites, we illustrate the effectiveness of HCHO data from geostationary measurements for understanding and predicting tropospheric ozone and its precursors.

## 1. Introduction

Formaldehyde (HCHO) is directly released to the atmosphere from sources that include motor vehicles, industrial activities, prescribed burnings, and wildfires. HCHO is one of the Hazardous Atmospheric Pollutants (HAP) – that are harmful to human health – defined by the US Environmental Protection Agency (see US EPA 2015a for more information). More importantly, HCHO is chemically produced during volatile organic compound (VOC) oxidation processes (Wolfe et al., 2016), and is therefore correlated with major chemical species formed during photochemical smog episodes [e.g., ozone ($O_3$)]. Because of the close relationship between HCHO and its VOC precursors, the ratio of satellite HCHO columns to nitrogen dioxide ($NO_2$) columns has been suggested as an indicator of photochemical regimes, i.e., the ratio determines "VOC-limited (or sensitive)" or "$NO_x$-limited (or sensitive)" regimes of $O_3$ formation in a certain location and season (Martin et al., 2004, Jin et al., 2017). In the presence of $NO_x$, HCHO can be a major source of hydroxyl radical (OH), the most important chemical species in the troposphere initiating photochemical chain reactions. The chemical lifetime of HCHO with respect to loss by OH reaction and photolysis is several hours (Warneke et al., 2011). HCHO is highly soluble and may contribute to aqueous chemical processes in clouds and precipitation in the atmosphere and in bodies of water at the Earth's surface (Barth et al., 2007.; Luecken et al.

2012).

Due to its importance to tropospheric chemistry, atmospheric chemists and the environmental remote sensing community have sought to produce high quality tropospheric HCHO retrievals. Because of its weak absorption in the ultraviolet (UV) spectral region, HCHO is regarded as one of the most difficult species to retrieve from satellite-based radiance observations in the UV-visible (UV-VIS) spectral region (e.g., GOME/GOME-2, SCIAMACHY, OMI, and OMPS; see Martin et al., 2003, Zhu et al., 2016 for references). In addition, the large uncertainties in satellite trace gas retrievals based on UV-VIS spectral measurements arise from the calculation of the air mass factor (AMF), which converts the slant column density of a trace gas to its vertical column values by considering the vertical sensitivity of the observations (AMF = slant column/vertical column, Palmer et al., 2001; Boersma et al., 2004; Lorente et al., 2017). Therefore, it is important to identify factors affecting the accuracy of HCHO retrievals and to find a method to reduce these uncertainties.

Palmer et al. (2001) expressed the AMF as a vertical integral of the product of scattering weight functions and normalized vertical profile shapes of trace gases that vary with atmospheric heights. The scattering weight function can be pre-calculated in a look-up table using radiative transfer (RT) model simulations, while the *a priori* profiles are

generally derived from a three-dimensional chemical transport model. This formulation has been widely used to derive operational trace gas retrieval products (e.g., Gonzalez Abad et al., 2015, De Smedt et al., 2017).

In this study, we examine the role of trace gas vertical profile shapes on HCHO retrievals in the Los Angeles (LA) Basin megacity. The HCHO retrievals from existing polar-orbiting satellites were investigated and utilized in previous studies (e.g., Palmer et al., 2001; Millet et al., 2008; Stavrakou et al., 2015; Abad et al., 2015; Zhu et al., 2016); these studies focused on regions with large biogenic sources or showed large scale contrasts between land and ocean. Zhu et al. (2014) estimated the anthropogenic VOC emissions from large industrial complexes in Houston, Texas, by oversampling OMI HCHO columns. In the near future, HCHO retrievals will be available from both geostationary [e.g., TEMPO (Fishman et al., 2012; Zoogman et al., 2017), GEMS (Kim et al., 2012), Sentinel-4 (Ingmann et al., 2012, Veihelmann et al., 2015)] and polar-orbiting (e.g., TROPOMI, Veefkind et al., 2012) platforms with much finer temporal and spatial resolutions, enabling satellite-based air quality studies at sub-urban to urban scales. HCHO retrievals at these scales may need a better strategy to deal with spatial and temporal variability in *a priori* vertical profiles of measured tracers than current methods that rely on profile shapes generated by coarse (horizontal grid resolutions of 1-3 degrees) global models. For

example, Heckel et al. (2011) investigated the impacts of the spatial resolution of *a priori*

profiles on $NO_2$ retrievals in a coastal city (San Francisco, California), which highlighted

the need for high resolution *a priori* data to quantitatively probe tropospheric pollution in

coastal regions and near localized sources such as power plants. Russell et al. (2011) also

5     found non-negligible impacts of high spatial and temporal resolution terrain and profile

inputs on the Ozone Monitoring Instrument (OMI) $NO_2$ retrievals. Kwon et al. (2017)

emphasized the importance of using hourly varying HCHO AMF for geostationary satellite

measurements in East Asia mainly due to temporal changes in aerosol chemical

composition and vertical distributions.

10        In this study, we simulate fine-resolution (4 km x 4 km) vertical profiles for HCHO

retrievals, and investigate the spatiotemporal variability of the HCHO AMF based on these

profiles. We also show the usefulness of detailed spatial and temporal information on

HCHO plume structures at an urban scale for interpreting the effectiveness of ozone

pollution controls.

**2. Data and models**

**2.1.  Aircraft and ground-based measurements**

  -  *NOAA WP-3 aircraft observations*

During the California Nexus of Air Quality and Climate Change (CalNex) campaign, the

NOAA WP-3 aircraft performed 20 research flights mainly over the LA Basin and the

Central Valley in California during May and June 2010 (see Ryerson et al., 2013 for more

information). The main goals of CalNex were to quantify the emissions of greenhouse

gases and ozone and aerosol precursors and to understand the chemical transformations

and the transport of pollutants. The NOAA WP-3 aircraft was equipped with a large suite

of gas phase and aerosol measurements. In this study, we use the HCHO measurement of

a Proton-Transfer-Reaction Mass-Spectrometry (PTR-MS) instrument onboard the WP-3

aircraft (Warneke et al., 2011). Airborne HCHO measurements by PTR-MS are difficult

due to a strong humidity dependency. The detection limit for HCHO with this instrument

is between 100 pptv in the dry free troposphere and 300 pptv in the humid marine boundary

layer. The PTR-MS HCHO measurements have been shown to agree with Differential

Optical Absorption Spectroscopy (DOAS) observations (Stutz and Platt, 1997; Platt and

Stutz, 2008) within the stated uncertainties. For comparison, the model results are first

sampled at the times and locations of the observations. Then the PTR-MS measurement

data onboard the P3 aircraft and the sampled model data are averaged at the model spatial

resolution (horizontal and vertical) to allow one-to-one comparison of the observations and

model results.

20-   *UCLA long-path DOAS data in Pasadena during CalNex*

UCLA's long-path (LP) DOAS instrument (Stutz and Platt, 1997; Platt and Stutz, 2008) is located on the California Institute of Technology (Caltech) campus on the roof of the Millikan Library at 35 m AGL (above ground level). Four retro-reflectors are situated northeast of the main instrument in the mountains behind Altadena at 78, 121, 255, and

556 m AGL. The average distance between the LP-DOAS telescope and the reflectors is about 6 km. Spectral retrievals of HCHO mixing ratios were performed in the 324-346 nm wavelength range using a combination of a linear and non-linear least squares fit, as described in Stutz and Platt (1996) and Alicke et al. (2002). Spectral absorption features of $O_3$, $NO_2$, HONO, $O_4$, and HCHO were incorporated in the fitting procedure. The campaign-

averaged statistical HCHO error in the DOAS measurements during CalNex was about 150 pptv (Warneke et al, 2011). In the present study, we use these ground-based DOAS data since vertical distribution information resulting from the four retroreflectors at different altitudes allows for comparison with the model results. The LP-DOAS data are averaged over the upper light path from 35 m AGL (Millikan Library at Caltech) to 225 m AGL

(water tank in Altadena) and are averaged for one hour prior to the comparison with the model results. The model values on the vertical levels corresponding to 35 m to 225 m AGL are averaged for the comparison with the LP-DOAS data. The model value from the 4 km x 4 km horizontal grid cell containing Millikan Library at Caltech is selected for the comparison with the LP-DOAS observations.

*The AQMD surface monitoring data*

The hourly $O_3$ data from the South Coast Air Quality Management District (AQMD) monitoring network (http://www.arb.ca.gov/aqmis2/aqdselect.php) are utilized for the trend study. Details on standard procedures for maintaining and operating air monitoring

stations and specific instrumentations are provided in the CARB air monitoring web manual (http://www.arb.ca.gov/airwebmanual/index.php). The locations of the sites and the data are shown in Auxiliary Material in Kim et al. (2016).

## 2.2. WRF-Chem model

We use version 3.4.1 of the Weather Research and Forecasting-Chemistry model (WRF-Chem, Grell et al., 2005). The model physical and chemical settings are the same as that used by Kim et al. (2016). The mother and the nested domains of the WRF-Chem model are the western U.S. (12 km x 12 km horizontal resolution) and the state of California (4 km x 4 km horizontal resolution), respectively. The model has 60 vertical levels with ~50

m thickness between vertical levels up to 4 km above ground level, with coarser vertical resolution at higher levels. The first model level where mixing ratios of chemical species are calculated is ~25 m. The simulation period is 26 April 2010 – 17 July 2010. Meteorological initial and boundary conditions are based on NCEP Global Forecast System data. The MOZART (Model for OZone And Related chemical Tracers,

http://www.acom.ucar.edu/wrf-chem/mozart.shtml) (Emmons et al., 2010) global model

results are used as initial and boundary conditions for the mother domain of WRF-Chem. Biogenic emissions are based on the Biogenic Emissions Inventory System (BEIS) version 3.13, with additional emissions from urban vegetation (Scott and Benjamin, 2003) are added. The Noah land surface model, YSU planetary boundary layer model, Lin

microphysics scheme, and Grell-Devenyi ensemble cumulus parameterization (only for the mother domain) are adopted (see references in Kim et al., 2009). The chemical mechanism is based on the Regional Atmospheric Chemistry Mechanism (RACM) (Stockwell et al., 1997) with ~30 reaction rate coefficients updated (Kim et al., 2009).

We adopt the $NO_x$ and CO emission estimates from Kim et al. (2016) that utilized

the fuel-based approaches of McDonald et al. (2012; 2013; 2014). For VOC emissions, we used the emission estimates from the top-down approach employing ground-based observations in Pasadena, as described by Borbon et al. (2013), along with the US EPA NEI05 (US EPA 2008; Kim et al., 2011; 2016) and NEI11 (US EPA 2015b; Ahmadov et al., 2015) inventories. The HCHO model results using the top-down VOC emissions

approach are the focus of this manuscript.

**2.3. VLIDORT radiative transfer model**

We used the Vector Linearized Discrete Ordinate Radiative Transfer (VLIDORT) model (Spurr, 2008) to calculate a trace gas AMF by vertically integrating the product of the

scattering weight function and the normalized vertical profile function of the trace gas, as

described by Palmer et al. (2001). VLIDORT is a multiple-scattering discrete ordinates RT model for stratified atmospheres. It applies the pseudo-spherical approximation to solve for the multiple scattering of photons in a stratified atmosphere; diffuse scattering is evaluated in a plane-parallel medium, but solar attenuation is performed in a spherical

atmosphere. Solar photon single scattering and viewing paths are treated precisely in a spherically curved atmosphere. Since VLIDORT is linearized, simultaneous generation of any number of analytically derived Jacobians with respect to profile quantities, column quantities, or surface properties is possible. We adopt the spectral resolution of 0.2 nm and a spectral range of 300.5-365.5 nm for our HCHO retrievals. The AMF presented in the

manuscript is selected at 340 nm, similar to the current satellite retrieval.   Solar zenith angles are 52.8°, 16.7°, and 28.8° at 16, 19, 22 UTC, respectively. Relative azimuth angles are 56.6°, 15.5°, 246.1° at 16, 19, 22 UTC, respectively. The viewing zenith angle in VLIDORT is 46.5°. We assume a constant surface reflectance of 0.05 across the domain. For snow-covered mountain top and desert areas, the surface reflectivity can be larger than

0.05, which would increase the sensitivity of satellite HCHO observations to the surface, and in turn would increase the AMF and further modify the spatial distribution of AMF in Southern California. The sensitivity of the HCHO AMF to the surface reflectivity for this area needs to be pursued in future study using data adequate for the TEMPO HCHO retrieval. Vertical profiles of HCHO, $O_3$, $NO_2$, $SO_2$, and BrO mixing ratios were used as

inputs to the VLIDORT simulations. We used the WRF-Chem model described above to

generate profiles of HCHO, $O_3$, $NO_2$ and $SO_2$, while for BrO, GEOS-Chem global model results were utilized.

## 3. Results

### 3.1. Observed and simulated HCHO

In order to use the model HCHO profiles for AMF calculations and to explore impacts of fine-resolution *a priori* on the retrievals, they should be reasonably good representations of the real atmospheric profiles. Therefore, we evaluate WRF-Chem HCHO simulations with the ground-based LP-DOAS data and aircraft PTR-MS observations. Figure 1 shows diurnal variations of the near-surface LP-DOAS HCHO observations and model results using various emission inventories on weekdays and weekends. The model results using either the top-down VOC emission estimates based on Borbon et al. (2013; red lines) and the NEI05 (Kim et al., 2016; blue lines) agree with the observations best. The model underestimates the LP-DOAS HCHO observations when we ignore the biogenic VOC emissions or adopt the most-up-to-date VOC inventory for year 2010 (NEI11, described in Ahmadov et al., 2015), with its lower anthropogenic alkene emissions than those from the NEI05 and top-down approaches. Maximum observed and modeled HCHO mixing ratios in Pasadena are about 4 ppbv during weekdays or 5 ppbv during weekends. During the weekends, faster photochemistry due to lower $NO_x$ emissions causes higher ozone and HCHO mixing ratios (Pollack et al., 2012; Kim et al., 2016).

Figure 2 shows the vertical profiles of potential temperature and HCHO mixing ratio from the aircraft observations and model results in the LA Basin on May 4, 2010. The potential temperature profiles in the model agree with the observations and help to characterize different vertical mixing regimes: a stable boundary layer near Catalina Island and the growth of the convective boundary layer from the LA urban cores eastward to the desert on the east side of the Basin. Similarly, the WRF-Chem HCHO profiles are in good agreement with the WP-3 PTR-MS observations. The convective boundary layer develops mainly by buoyancy forcing during daytime and leads to well-defined boundary layer heights (or mixing heights) ranging from a few hundred meters to several kilometers and well-mixed vertical profiles of potential temperature and scalars. Meanwhile, stable boundary layers are characterized by a shallow boundary layer (boundary layer height of maximum a few hundred meters), a positive vertical gradient of potential temperature near the surface, and poorly-mixed vertical profiles of scalars because of weak turbulent mixing that frequently occurs over the ocean or during nighttime. Overall, our model results agree with the observations from the aircraft and ground-based observations; therefore, it is reasonable to use the model HCHO profiles as inputs to VLIDORT and to examine the AMF results from this RT model.

**3.2. Spatial distribution of AMF and sensitivity to *a priori* profiles at different times of day**

The spatial distribution of the VLIDORT HCHO AMF using the WRF-Chem profiles at 4 km x 4 km resolution at different times of day on May 4, 2010 is shown in Figure 3. The AMF ranges from 0.6 to 1.2 within the LA Basin and in the nearby coastal areas. The AMF values are 0.6-0.7 in the urban cores. In contrast, for high mountains such as the Los Padres

Forest located in the northwestern part of the Basin, the AMF is greater than 1. Above the Pacific Ocean near the coast, the AMF is about 0.9-1. These results are similar to the AMF calculations by Palmer et al. (2001); they obtained AMF = 0.71 in Tennessee, where high isoprene levels are seen in the boundary layer, and AMF = 1.1 over the North Pacific. The AMF values calculated by Palmer et al. (2001) resulted in Global Ozone Monitoring

Experiment (GOME) measurements that were ~35% less sensitive to the HCHO column (or 35% smaller total AMF) over Tennessee than over the North Pacific. Palmer et al. (2001) also noted small AMF values over California, which they attributed to a shallow boundary layer resulting from strong subtropical subsidence combined with a strong surface source of HCHO from biogenic hydrocarbons. Our study agrees with this finding, except that both

anthropogenic and biogenic VOC contribute to high formaldehyde in the LA Basin (Figure 1). General features of the AMF distribution in the area do not change significantly when a constant surface pressure is used in the RT simulations (see Supplementary Material Figure S1 and S2). 82% (99%) of the area shows the differences of AMF less than 5% (10%). The direct influence of complex terrain height on the AMF is small. Similarly, the

spatial pattern was not strongly affected by the currently available bottom-up emission

inventory used to generate the WRF-Chem HCHO profiles in our study (see Supplementary Material Figure S1 and S2). 95% (98%) of the area shows the differences of AMF less than 5% (10%). The impact of bottom-up emission inventory was larger in Barkley et al. (2012) when various isoprene emission inventory over tropical South

America were included in the satellite HCHO retrievals: in general, the difference in the HCHO columns was ±20% and for individual locations, it was up to ±45%. Thus, the role the bottom-up emission inventory play in the AMF calculation varies depending on the quality (accuracy) of the emission inventories and their impacts on the profile shapes.

As mentioned above, the most operational HCHO retrievals adopted global model

results at roughly 1°-3° grid size as a priori profile, which are ~1000 times as large as the spatial resolution in our study (4 km x 4 km). For the domain of interest in this study, the global model has just a few profiles. Here we compare the AMF from global model results (2° latitude x 2.5° longitude resolution) as *a priori* in the Smithsonian Astrophysical Observatory (SAO) OMI formaldehyde retrieval (Gonzalez Abad et al., 2015) with the

AMF from this study for the LA basin and discuss more on the spatial resolution effect. In contrast to the AMF in this study as in Figure 4, the AMF in SAO OMI formaldehyde retrieval does not vary much in the basin and is close to 1 (see Figure S3 in Supporting Material for details). The average of AMF from the OMI SAO product for the domain (33.5N-34.5N, 117W-118.5W) is 1.12 while the same domain average of AMF from this

study is 0.76. If AMF in this study is used, the HCHO column can increase by 47% on the

domain-average (up to ~100% at a finer scale), compared with the OMI HCHO column. The vertical HCHO profile in the OMI SAO product is almost a constant in the domain while the model profile at 4 km x 4 km resolution varies substantially. We will discuss the spatial resolution effect on the intensity of HCHO plumes in depth later.

Geostationary satellites such as TEMPO (Fishman et al., 2012; Zoogman et al., 2017), GEMS (Kim et al., 2012), and Sentinel-4 (Ingmann et al., 2012; Veihelmann et al., 2015) are expected to provide diurnally varying information about tropospheric pollution during daytime. It is, therefore, useful to investigate if diurnally varying *a priori* profile information is needed for accurate retrievals of satellite-based HCHO columns. Figure 3

shows the spatial distribution of VLIDORT HCHO AMF using the WRF-Chem profiles at 16, 19, and 22 UTC (equivalent to 9, 12, 15 Pacific Daylight Time, PDT, respectively) and HCHO columns. Overall, similar patterns of the AMF distribution are shown at all times: low AMFs in the urban cores and high AMFs in the area of Los Padres National Forest located in the northwestern region of the Basin. However, there are noticeable diurnal

changes in the AMFs over the high terrain east and northeast of downtown LA and over the Pacific Ocean near the coast, due to changing photochemical production and destruction and transport of HCHO throughout the day (Figure 3). Overall, minimum AMF values are reduced between morning and afternoon as HCHO is photochemically produced. At 15 PDT, AMF values < 0.6 (the white shading in Figure 3) occur in the mountainous

regions, including the San Gabriel Mountains, San Bernardino National Forest, Mt. San Jacinto, and Anza-Borrego Desert State Park. Onshore transport of photochemically produced HCHO plumes from downtown LA to the mountains occurs in the afternoon (see HCHO columns in Figure 3).

Figure 4 shows vertical distributions of the model HCHO mixing ratios at several locations in the LA Basin and the Pacific Ocean for the AMF values at different times of day [see Figure S4 in Supporting Material for the plots with number density unit (molecules cm$^{-3}$)]. Over the Pacific Ocean, the HCHO mixing ratio is small near the surface and more abundant at higher altitudes. The AMF over the ocean increases with time from 0.86 at 09

PDT to 1.03 at 15 PDT as the HCHO mixing ratio decreases with time, probably due to transport of the plume from the ocean to the inland area (see Supporting Material Figure S5 for detailed analyses). Over the land, the HCHO mixing ratio is higher in the boundary layer than in the free atmosphere. In the Los Padres National Forest where the highest AMF (0.91-1.21) occurs, the boundary layer grows with time, but the mixing ratio of HCHO is

small (< 1 ppbv). In Pasadena and at the LA Main St. site, the boundary layer heights and HCHO mixing ratios increase from 9 PDT to 12 PDT. The maximum HCHO value in the boundary layer is about 6 ppbv. The HCHO in the boundary layer decreases at 15 PDT, but mixing ratios above the boundary layer (> 1 km) increase due to the upper level easterly

transport of the HCHO plumes. Consequently, the AMF decreases from 0.7 at 9 PDT to 0.6 at 12 PDT and then increases to 0.7 at 15 PDT, due to an enhanced sensitivity to increased upper-level HCHO mixing ratios. For these urban core sites, the HCHO AMF ranges from 0.6 to 0.7. In the San Gabriel Mountains, San Bernardino National Forest, and Mt. San

Jacinto, the boundary layer height is well defined and shallow and does not change significantly throughout the day. However, the AMF values change substantially (decreasing by ~40%) throughout the day over these locations; this is likely because HCHO mixing ratios increase between morning and afternoon, mainly due to transport and formation of the plumes originating from urban core regions. The AMF at Anza-Borrego

Desert State Park decreases with time from 0.96 at 9 PDT to 0.71 at 15 PDT due to increasing HCHO mixing ratios, in spite of the increase in boundary layer height. These findings highlight that the importance of using time-varying, high spatial resolution *a priori* profile information for the accurate retrieval of geostationary HCHO measurements.

We extended this analysis in Figure 5, where for ranges of HCHO AMF (e.g., 1.0

< AMF < 1.1) across the model domain, the model HCHO profiles are averaged and plotted at the three times (9, 12, 15 PDT) [see Figure S6 in Supporting Material for the plots with number density unit (molecules $cm^{-3}$)]. Each plot shows that the AMF values are smaller when the HCHO mixing ratios are higher near the surface. At 12 and 15 PDT, as expected,

the profiles have more well-mixed shapes for deeper vertical layers. The dependence of the AMF value on the profile shape is similar at each time of day: the higher AMF is related to lower HCHO mixing ratios (or number densities) in the atmospheric boundary layer (up to 1-3 km altitude AGL). More quantitative analysis is shown below.

Using all available data points, we investigate the relationship between AMF and the HCHO mixing ratio at 200 m in the boundary layer at different times of day in Figure 6 [see Figure S7 in Supporting Material for the plots with number density unit (molecules $cm^{-3}$)]. The plot illustrates that as the HCHO mixing ratio increases, the AMF decreases. At all times investigated, AMF is anti-correlated with HCHO mixing ratio (or number

density). Correlation coefficients between AMF and HCHO mixing ratio are -0.68, -0.85 and -0.84 at 16 (09), 19 (12), and 22 (15) UTC (PDT). In general, AMF values decrease from morning to late afternoon. The AMF values are reduced substantially for HCHO mixing ratio of 2, 3, and 4 ppb. Therefore, it is useful to examine if the HCHO mixing ratios of 2, 3, and 4 ppb or higher can be captured at coarser spatial resolutions. Figure 7

demonstrates a scatter of HCHO concentrations at 4 km x 4 km resolution on a coarser grid from 8 km to 300 km. Here the values for coarse grids are generated from the spatial averages of the original model results at 4 km resolution in this study. A scatter of concentrations is getting larger at a spatial grid size $\geq$ 20 km. For example, the concentration at 4 km resolution varies from 1 to 6 ppb while that at 100 km resolution is

about 2 ppb. Table 1 summarizes the efficiency of capturing the plumes that have greater

HCHO mixing ratio than the reference values for each spatial grid resolution. Of particular

importance are the reference values of 2, 3, 4 ppb for which AMF is greatly reduced. Table

1 indicates that the grid size ≤ 12 km can capture the plumes of HCHO VMR > 4 ppb or 5

ppb at 4 km by more than 70%. If the grid size is 8 km, the plumes of 1-5 ppb are detected

5    by ~80%. If the grid size is greater than 100 km, it does not capture the plume of VMR >

2 ppb at this urban location. Thus, the AMF using the coarse resolution ≥ 100 km is about

because of low concentration < 2 ppb. Currently typical spatial resolution of regional-

scale models for the viewing domain of the geostationary satellites (e.g., air quality forecast

models for the U.S.) is 12-30 km in each latitude and longitude direction. Our

recommendation is to select the resolution as close as 4 km. Since the model simulation at

4km resolution is computationally expensive for the current geostationary satellite viewing

domain and all of high quality input data to the model are not readily available at this

resolution (e.g., emission inventory), the model simulations at 8-12 km resolution are

recommended to test and improve the model simulations and finally acquire *a priori* profile

for next generation environmental geostationary satellite retrievals if computing resources

are available.

    For UV-VIS retrievals, it is well known that the vertical profile shape affects the

value of the AMF. Our study suggests a strong anti-correlation between the absolute

concentration and the AMF: the AMF is low in the area of intense HCHO plumes. The

changes in the absolute HCHO concentrations in the boundary layer (altitude AGL < 1-3

km) strongly modify profile shapes, which in turn affect AMF substantially. To understand

the importance of the absolute magnitude of HCHO mixing ratios within the context of the

mathematical formula of AMF used, we examine shape factor, scattering weight function,

and AMF quantitatively. According to Palmer et al. (2001), AMF is expressed as

$$AMF = AMF_G \int_0^\infty w(z)S_z(z)dz. \qquad (1)$$

Here $AMF_G$ is a geometric air mass factor that is a function of solar zenith angle and

satellite viewing angle, $w(z)$ is a scattering weight that is associated with the sensitivity

10   of the backscattered spectrum to the abundance of the absorber at altitude z, and $S_z(z)$ is

a vertical shape factor for the absorber representing a normalized vertical profile of number

density. The vertical shape factor is defined as

$$S_z(z) = \frac{n(z)}{\Omega_v} \qquad (2)$$

, where $n(z)$ is the number density (molecules cm$^{-3}$) at altitude z and $\Omega_v$ is the vertical

column density or column (molecules cm$^{-2}$) of HCHO. In this manuscript, AMF in

Equation (1) is vertically integrated to the top of model domain that is roughly the top of

troposphere or above. Therefore, AMF here is tropospheric AMF. To understand the sensitivity of AMF on the vertical distribution, we also define $\Delta AMF_i$, a discrete increment of AMF for each model layer.

$$\Delta AMF_i = AMF_G \, w_i \, S_{zi} \, \Delta z_i \tag{3}$$

, where $i$ is an index representing the vertical grids, $\Delta z_i$ is the layer depth for the grid $i$, and $\sum \Delta AMF_i = AMF$.

In Figure 8, the vertical shape factor in Equation (2), the scattering weight

10 (multiplied by geometric AMF), and $\Delta AMF_i$ are plotted as a function of height over the North Pacific Ocean, San Gabriel Mountains, and Anza Borrego Desert State Park at 16, 19, and 22 UTC (see Figure 4 for the locations of these sites). The differences in the shape factor over the North Pacific Ocean are clear at altitudes $> \sim 1000$ m: the shape factor values at 22 UTC are larger than those at 16 and 19 UTC. In contrast, the HCHO column at 22

15 UTC is smaller than those at 16 and 19 UTC over the ocean (Figure 4). As the column density value decreases, the shape factor above ~1000 m becomes larger and causes higher $\Delta AMF_i$ and (tropospheric) AMFs, because a column density value is used as a normalization parameter for a shape factor. In order words, the satellite measurement is

more sensitive to the profile at 22 UTC than that at 16 UTC at this point over the Pacific Ocean.

For the San Gabriel Mountain site, the HCHO is confined below ~1400 m at 16, 19, and 22 UTC (there are no significant changes in boundary layer height during this time period) and its mixing ratio increases with time (Figure 4). The shape factor at 19 and 22 UTC is higher than that at 16 UTC below ~1400 m altitude (Figure 8, middle row). However, above this height, the shape factor and $\Delta AMF_i$ decrease with time: both are largest at 16 UTC and smallest at 22 UTC. The tropospheric AMF follows $\Delta AMF_i$ above ~1400 m and also decrease with time from 1 to 0.58. Thus, the satellite measurement is more sensitive to the profile at 16 UTC than that at 22 UTC in this mountainous area. The plot over the San Gabriel Mountain area illustrates that not only boundary layer height, but also the absolute magnitude of HCHO, influence the AMF value.

Anza-Borrego Desert State Park represents an example of a case in which both boundary layer height and HCHO mixing ratio increase with time (Figure 4 and Figure 8, bottom row). In case of the lowest boundary layer height (at 16 UTC), AMF is largest (AMF=0.98). When the boundary layer height is the highest among the three time periods (at 22 UTC), the AMF is smallest (AMF=0.71). For Anza Borrego Desert State Park, total column or near surface HCHO mixing ratio affect the shape factor, which in turn leads to

an AMF that is inversely proportional to the total column or near surface HCHO mixing

ratio. As shown in Figure 8, the shape factor and $\Delta AMF_i$ above the boundary layer

decrease with time, which causes a decrease in the tropospheric AMF with time.

In summary, the absolute value of the column or near-surface mixing ratio of

HCHO affects the shape factor as a normalization factor, in particular the value in the free

troposphere (above boundary layer), which dominates the tropospheric AMF. When the

HCHO mixing ratio is low in the boundary layer, the relative importance of the absorber

in the free troposphere increases. Conversely, when the HCHO mixing ratio is high in the

boundary layer, the relative importance of absorber in the free atmosphere decreases. Our

result suggests that a representation of the HCHO AMF using accurate fine-resolution *a*

*priori* profile information is critical to identify HCHO plumes and to place better

constraints on VOC emissions.

Although the focus of this manuscript is on the shape factor, we also investigate the

impacts of aerosol loading on AMF for the 8 sites shown in Figure 4. When the aerosol

optical properties from the model results are incorporated in our RT model calculations,

the AMF is reduced by ~10% at the N. Main St. and Pasadena sites and by < 10% at other

sites (Table 2). The aerosol optical depth, single scattering albedo, and asymmetry factor

calculated from the model results for the 8 sites are about 0.5, 0.9, and 0.7, which is close

to the values suggested as most probable atmospheric conditions in the LA Basin (see Table

in Baidar et al., 2013). Because the model aerosol results were not thoroughly evaluated

and optimized and only 8 sites were tested, the analysis of aerosol impact in this study is

limited. It is possible that some of the simulated aerosol components are overestimated,

because the emission inventory is not fully up to date for primary aerosol emissions and

aerosol precursor gases (e.g., overestimations of black carbon and $SO_2$ by a least a factor

of 3). Meanwhile, the AMF changes from the values at 16 UTC (09PT) due to diurnal

variations in *a priori* profile shape range from -40% to 20% (Table 2). It is likely that the

impact of aerosols on the AMF is relatively small when compared with the impact of the

profile shape factor examined in this study for the LA basin. De Smedt et al. (2015) and

Wang et al. (2017) also reported the importance of *a piori* profile shapes for an

improvement of satellite-based HCHO retrievals in Beijing, Xianghe, Wuxi in China.

Kwon et al. (2017) demonstrated that the impact of aerosol loading on HCHO AMF can be

large over East Asia, in particular, for a case of Asian dust transport in contrast to our study

for the LA basin.

**3.3. Air quality application of fine-resolution geostationary HCHO columns**

In this section, we illustrate the application of future geostationary HCHO retrievals to the

study of air quality, by using the WRF-Chem HCHO columns as a proxy for satellite data.

Figure 9 demonstrates the distribution of the ratio of HCHO to $NO_2$ tropospheric vertical

columns from the WRF-Chem model in the LA Basin at different times of day and on

weekdays and weekends for May-June 2010. For more information about the model $NO_2$

columns, refer to Kim et al. (2016).

Ratios of HCHO to $NO_2$ columns provide critical information about chemical

regimes relevant to controlling ozone pollution (Martin et al., 2004; Jin et al., 2017). In

Figure 9, the light blue to blue contours (HCHO/NO2 < 1) represent VOC-sensitive (or

VOC-limited) ozone production regimes, while the pink to the red contours (HCHO/$NO_2$

> 1) denote $NO_x$-sensitive regimes. During weekdays in 2010, most of the LA Basin is in

the VOC-sensitive regime, where a reduction in $NO_x$ emissions can cause an increase in

$O_3$. In the late afternoon during weekends, the broad polluted area becomes $NO_x$-sensitive,

so that $NO_x$ reductions lead to $O_3$ decreases.

Figure 10 shows 2000-2010 trends in surface $O_3$ from monitors in Pasadena and

San Bernardino. During this decade, $NO_x$ emissions were decreasing in the LA Basin,

largely due to better control of motor vehicle pollution (McDonald et al., 2012). On

weekdays during this decade, there was not a declining trend in surface $O_3$ in Pasadena,

while $O_3$ increased in San Bernardino. In contrast, on weekends, $O_3$ decreased between

2000 and 2010 in both Pasadena and San Bernardino. These observed $O_3$ trends are consistent with analyses of the ratio of HCHO to $NO_2$ columns, and their representation of VOC/$NO_x$ sensitivity, shown in Figure 10. Baidar et al. (2015) found that the spatial extent and the trend of higher $O_3$ during weekends than weekdays had decreased in the LA Basin

because of the increased tendency of lower $O_3$ during hot weekends, especially after the 2008 economic recession.

The polar-orbiting satellite instruments that are currently available do not provide diurnally varying information on HCHO/$NO_2$ columns and VOC/$NO_x$ sensitivities, because these measurements are made once a day in either the morning or early afternoon.

The discussion above makes it clear that future geostationary satellite HCHO and $NO_2$ columns will provide useful information about photochemical ozone regimes that could be used to evaluate pollution control policies.

**4. Summary and Conclusions**

Our tests of the sensitivity of HCHO AMF to several factors confirm the importance of *a priori* HCHO profile shapes. Our study reveals that the AMF is very sensitive to the absolute HCHO mixing ratio (or number density) in the boundary layer. Therefore, the absolute magnitude of HCHO concentration in the boundary layer is an essential factor in determining the AMF. For the coastal LA Basin megacity studied in this work, the AMF

values are inversely proportional to the magnitude of the HCHO mixing ratios in the boundary layer. Furthermore, the AMF over land is lower in the late afternoon (15 PDT) than in the morning (09 PDT), because of increasing HCHO mixing ratios throughout the day. Therefore, diurnal updates and fine spatial resolution *a priori* profile shapes are likely

to improve the retrievals of satellite-based HCHO columns.

The spatial distributions of fine-scale model HCHO columns in the LA Basin show hot spots in downtown LA around noon and enhancement and transport of the plumes to the eastern part of the Basin in the late afternoon. The ratio of HCHO to $NO_2$ columns during weekdays and weekends provides information on the chemical regimes relevant to

ozone formation at various locations and times in the Basin. Future geostationary satellites (e.g., TEMPO) may provide similar information, which could be used to assess the effectiveness of existing pollution controls and could help in planning or revising air pollution control policies.

**Acknowledgements**

The NOAA Health of Atmosphere program, the NASA ROSES ACMAP (NNH14AX01I), and NASA GEO-CAPE Mission Pre-formulation Study (NNH13AW31I) supported this research. The authors thank G. Gonzalez Abad and K. Chance at Harvard SAO and H.-J. Lee at Pusan National University.

The WRF-Chem model version 3.4.1 used in this study is available at http://www2.mmm.ucar.edu/wrf/users/download/get_source.html. We acknowledge use of MOZART-4 global model output available at http://www.acom.ucar.edu/wrf-chem/mozart.shtml. The CalNex field campaign data are available at

5    http://www.esrl.noaa.gov/csd/projects/calnex/.

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

**Figure captions**

Figure 1. Diurnal variations of ground-based observations (black filled circles with solid

lines) of HCHO and the corresponding model simulations (lines without symbols) in

Pasadena (34.1370°N, 118.1254°W) averaged for weekdays (left) and weekends (right)

during May-June 2010. All model simulations utilized the fuel-based $NO_x$ and CO

emissions in Kim et al. (2016). The red solid line shows the results utilizing the VOC

emissions from the top-down approach in Borbon et al. (2013), the red dashed line denotes

the same model settings represented by the red solid line (top-down approach) except for

zero biogenic VOC (BVOC) emissions, the blue solid line represents the model results

using the VOC emissions from NEI05 (as in Kim et al., 2016), and the light blue line shows

the model output using the VOC emissions from NEI11.

Figure 2. The flight path of NOAA WP-3 (top) and the spatial distribution of vertical

profiles of aircraft observed and model simulated potential temperature (middle) and

HCHO (bottom) in the LA Basin on May 4, 2010. The black filled circles and red solid

lines/symbols represent the observations and model results, respectively.

Figure 3. The spatial distributions of air mass factors from the radiative transfer model calculations (left) and HCHO columns (right) in the LA Basin at 16 UTC (top), 19 UTC (middle), and 22 UTC (bottom) on May 4, 2010. The black filled circles are included as points of further investigations, representing background, urban cores, and downwind

regions.

Figure 4. Vertical profiles of HCHO are shown for various points of interest (red symbols on a Google map). Blue, orange, and magenta lines represent 16, 19, and 22 UTC (or 09, 12, 15 PDT), respectively on May 4, 2010. The altitude AGL is shown. The same plot with

the unit of molecules $cm^{-3}$ is shown in the Supplementary Material.

Figure 5. Vertical profiles of HCHO averaged for the AMF value intervals (as in legends) at 16, 19, and 22 UTC (left to right) are displayed. Thick lines with symbols are for averages and thin dotted lines are for one standard deviation values. The altitude AGL is

shown. The same plot with the unit of molecules $cm^{-3}$ is shown in the Supplementary

Material.

Figure 6. The relationship between the AMF and model HCHO volume mixing ratio is demonstrated. Different colors denote different times. The HCHO mixing ratio at ~200 m altitude AGL is plotted. The same plot with the unit of molecules $cm^{-3}$ is shown in the Supplementary Material.

Figure 7. Comparison of HCHO mixing ratios at 4 km x 4 km resolution with mixing ratios at coarser resolutions of (a) 8 km x 8 km, (b) 12 km x 12 km, (c) 20 km x 20 km, (d) 36 km x 36 km, (e) 48 km x 48 km, (f) 100 km x 100 km, (g) 200 km x 200 km, and (h) 300 km x 300 km. One-to-one line is shown in black.

Figure 8. Vertical profiles of (left) shape factor and scattering weight and (right) $\Delta AMF_i$ (discrete increment of AMF) at North Pacific Ocean, San Gabriel Mountains, and Anza Borrego Desert State Park. Scattering weights multiplied by geometric AMF are shown. Dashed lines and solid lines with symbols in the left panel denote the scattering weight and shape factor, respectively. Total (or tropospheric) AMF values are shown in legends in the right panel.

Figure 9. Spatial distributions of the ratios of the model HCHO column to $NO_2$ column during weekdays (left) and weekends (right) at 09 PDT, 12 PDT, and 15 PDT for May-June 2010. The light pink to red colored contours denote the area under the $NO_x$-limited chemical regime.

Figure 10. Decadal $O_3$ trends in Pasadena and San Bernardino during weekdays (red) and weekends (blue) are shown. The linear least square fits of $O_3$ for Wednesday and Sunday are plotted in dashed lines.

**Table captions**

Table 1. Percentage (%) of intense HCHO plumes retained as the spatial resolution is

changed from 4 km. Each column shows the fraction of the plumes retained at coarser

resolutions. Here the plume is defined by the area in which the HCHO mixing ratio is

greater than the reference HCHO volume mixing ratio (VMR) (1-6 ppb) at 4 km resolution.

For example, the second column shows how much area at 8-200 km resolution has a HCHO

VMR > 1 ppb when compared with the area with VMR > 1 ppb at 4 km resolution.

Similarly, the last column shows how often a model HCHO VMR is greater than 6 ppb at

8-200 km resolution compared with the same plume of VMR > 6 ppb at 4 km resolution;

all coarser resolutions (8-200 km) fail to capture this most intense plume. Only model

HCHO results at 200 m above ground level at 19 UTC (12 PDT) are used. The areas with

HCHO VMRs greater than 1, 2, 3, 4, 5, or 6 ppb are 92800, 29136, 12832, 4256, 848, or

64 km$^2$, respectively in the original simulations at 4 km resolution. The area of the domain

is 143856 km$^2$.

Table 2. Summary of air mass factors at 8 locations at 16-22 UTC (09-15 PDT). The results

without/with aerosols impacts are also shown.

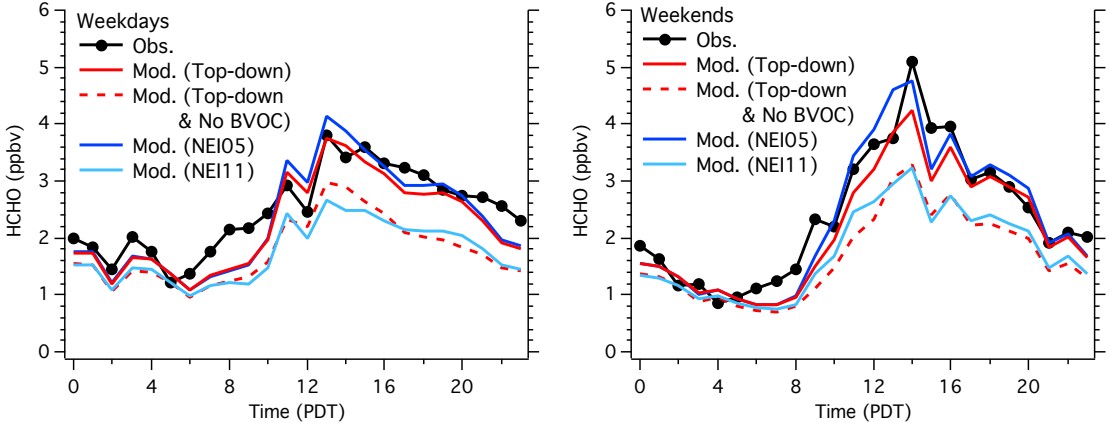

Figure 1. Diurnal variations of ground-based observations (black filled circles with solid lines) of HCHO mixing ratio and the corresponding model simulations (lines without symbols) in Pasadena (34.1370°N, 118.1254°W) averaged for weekdays (left) and weekends (right) during May-June 2010. All model simulations utilized the fuel-based $NO_x$ and CO emissions in Kim et al. (2016). The red solid line shows the results utilizing the VOC emissions from the top-down approach in Borbon et al. (2013), the red dashed line denotes the same model settings represented by the red solid line (top-down approach) except for zero biogenic VOC (BVOC) emissions, the blue solid line represents the model results using the VOC emissions from NEI05 (as in Kim et al., 2016), and the light blue line shows the model output using the VOC emissions from NEI11.

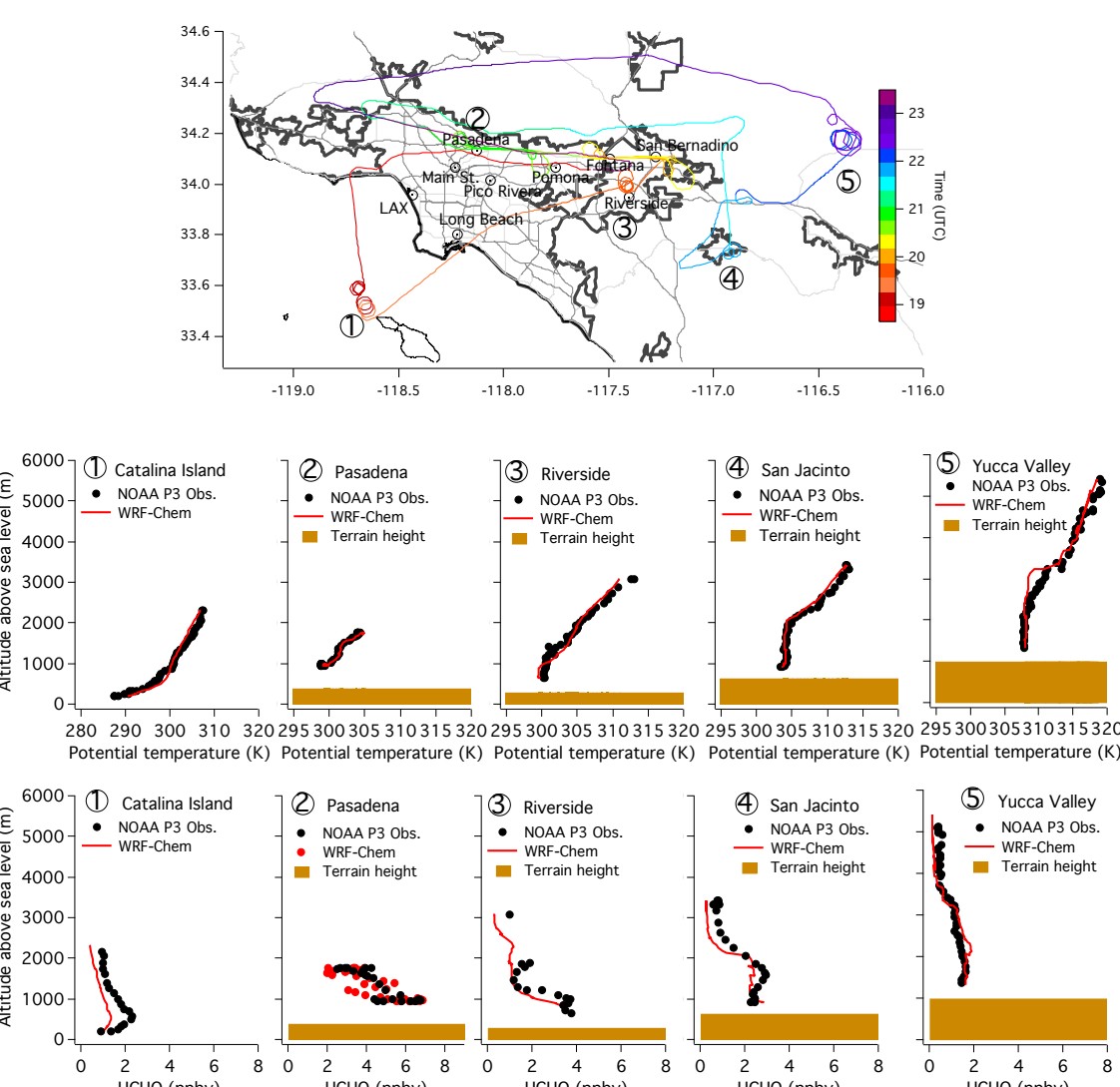

Figure 2. The flight path of the NOAA WP-3 aircraft (top) and the spatial distribution of vertical profiles observed on the aircraft and simulated by the model for potential temperature (middle) and HCHO (bottom) in the LA Basin on May 4, 2010. The black filled circles and red solid lines/symbols represent the observations and model results, respectively.

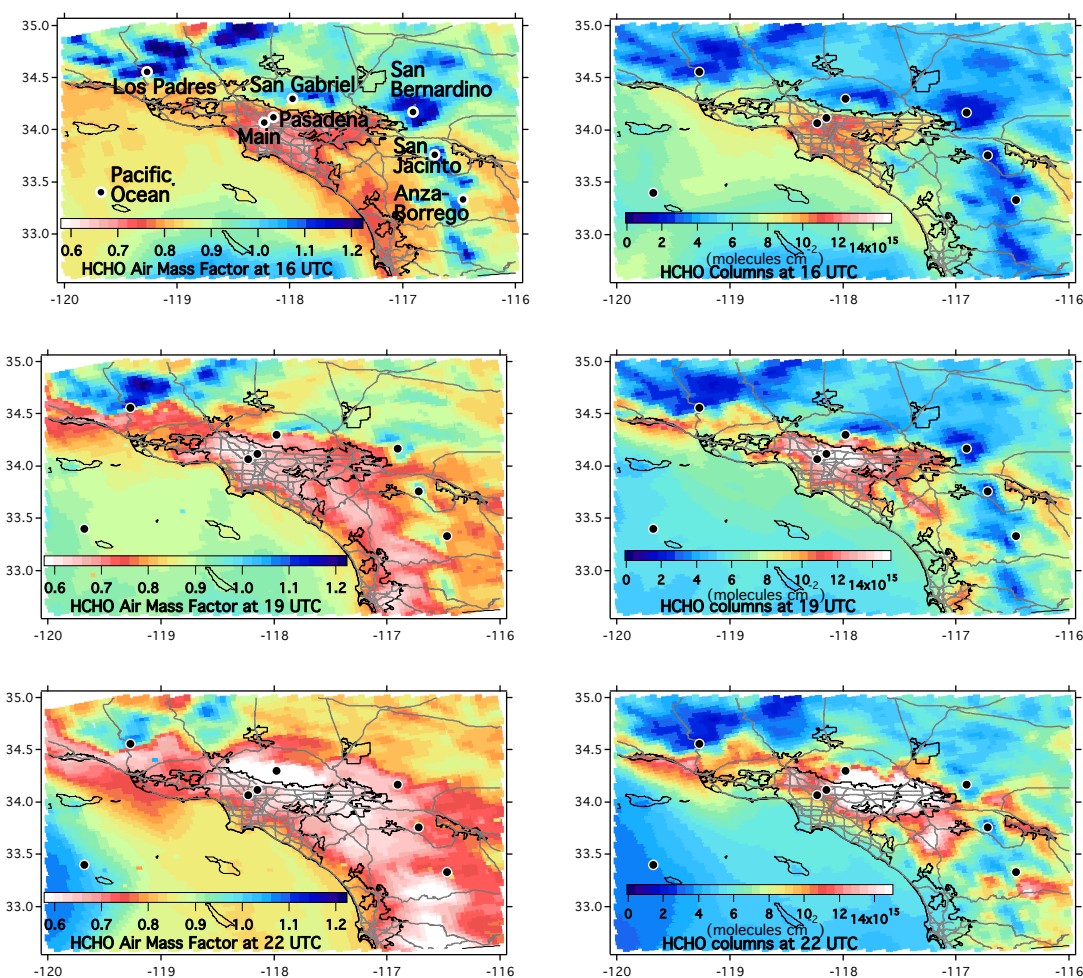

Figure 3. The spatial distributions of air mass factors from the radiative transfer model calculations (left) and HCHO columns (right) in the LA Basin at 16 UTC (top), 19 UTC (middle), and 22 UTC (bottom) on May 4, 2010. The black filled circles are included as points of further investigations, representing background, urban cores, and downwind regions.

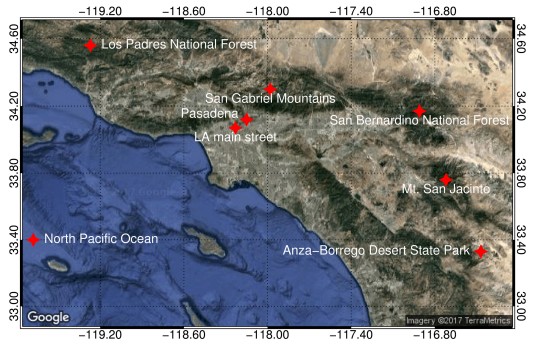

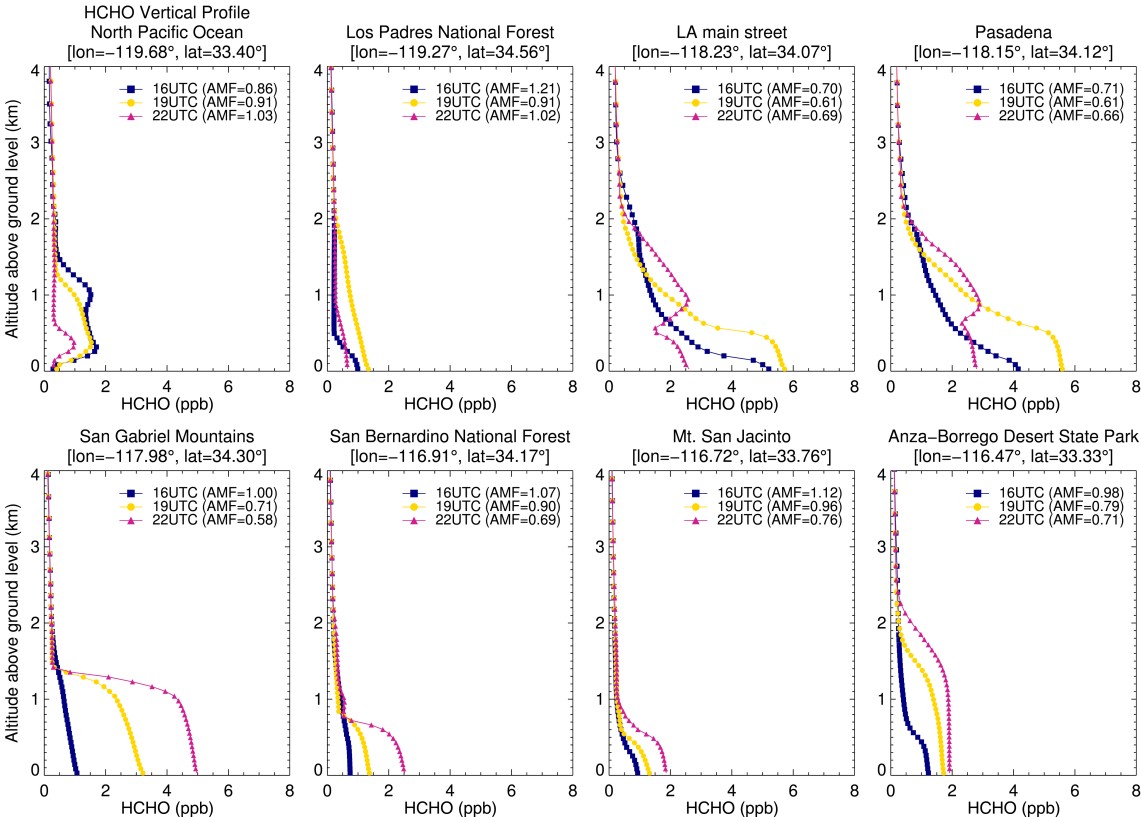

Figure 4. Vertical profiles of HCHO mixing ratio are shown for various points of interest (red symbols on a Google map). Blue, orange, and magenta lines represent 16, 19, and 22 UTC (or 09, 12, 15 PDT), respectively on May 4, 2010. The altitude AGL is shown. The same plot with the unit of molecules cm$^{-3}$ is shown in the Supplementary Material.

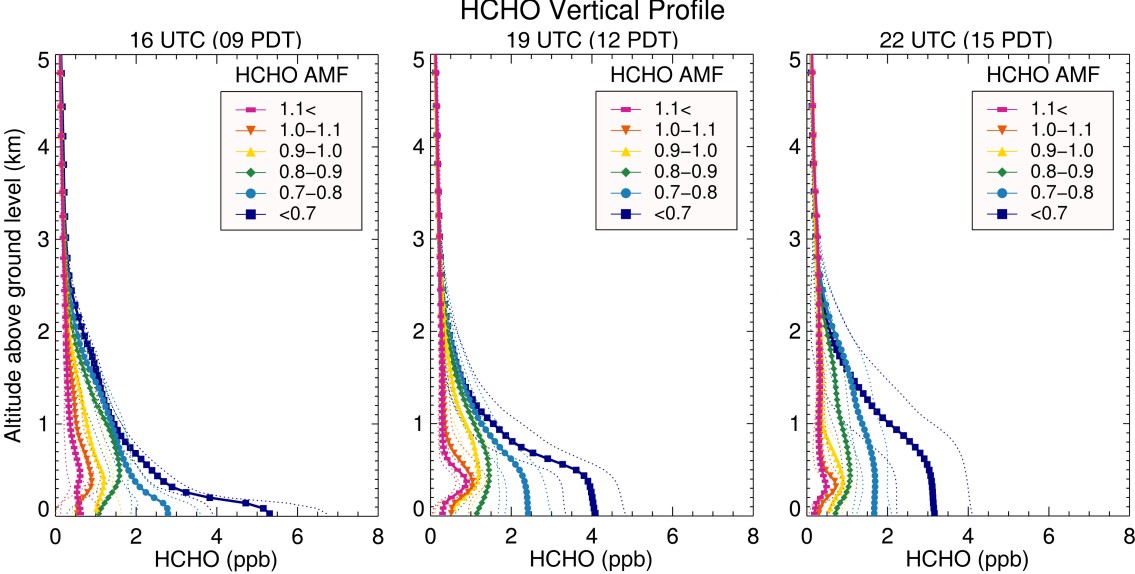

Figure 5. Vertical profiles of HCHO mixing ratio averaged for the AMF value intervals (as in legends) at 16, 19, and 22 UTC (left to right) are displayed. Thick lines with symbols are for averages and thin dotted lines are for one standard deviation values. The altitude AGL is shown. The same plot with the unit of molecules cm$^{-3}$ is shown in the Supplementary Material.

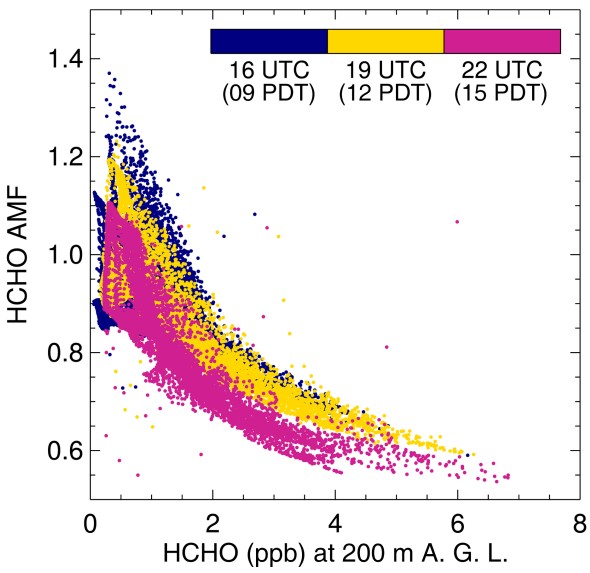

Figure 6. The relationship between the AMF and model HCHO volume mixing ratio is demonstrated. Different colors denote different times. The HCHO mixing ratio at ~200 m altitude AGL is plotted. The same plot with the unit of molecules cm$^{-3}$ is shown in the Supplementary Material.

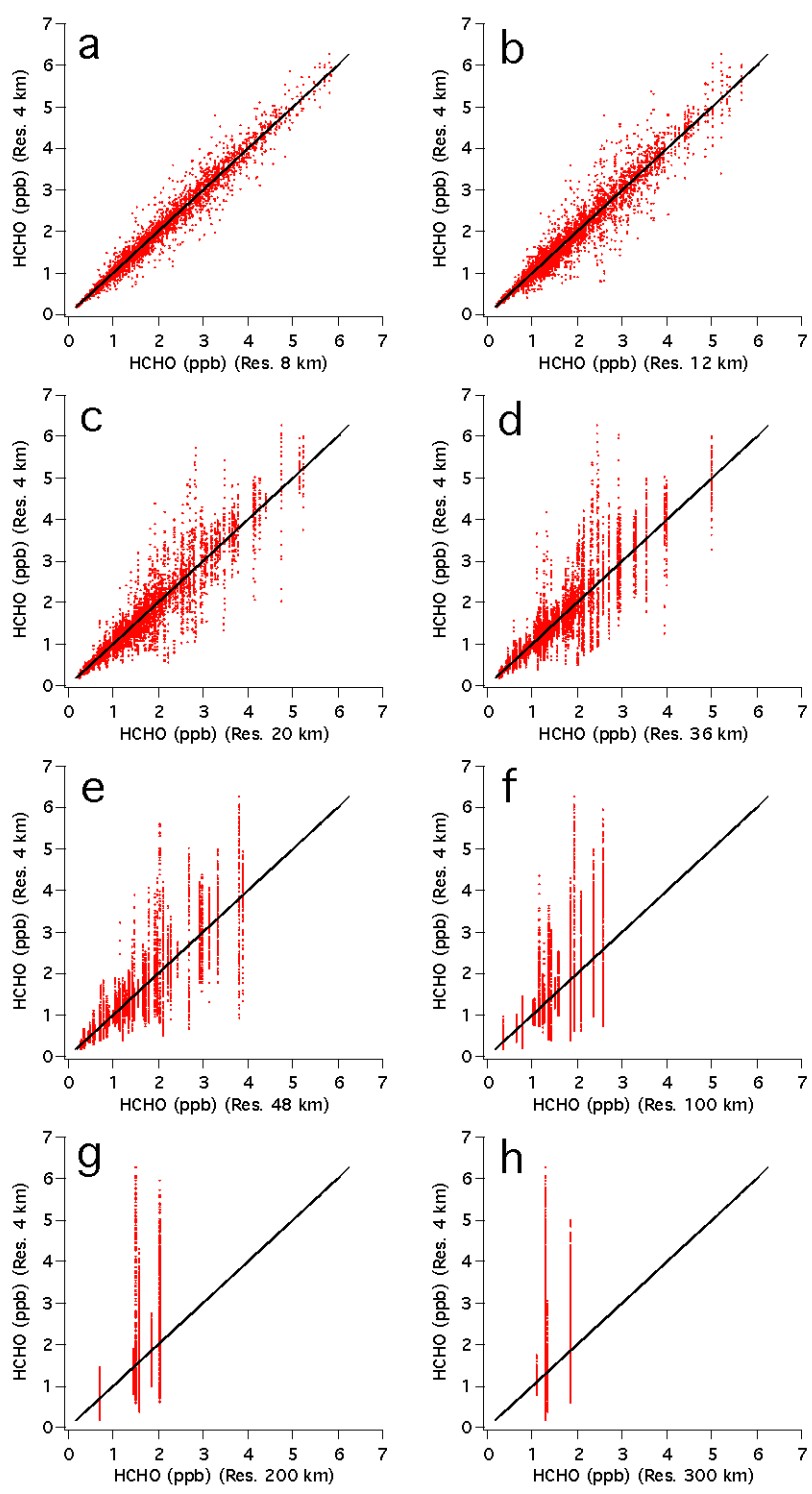

Figure 7. Comparison of HCHO mixing ratios at 4 km x 4 km resolution with mixing ratios at coarser resolutions of (a) 8 km x 8 km, (b) 12 km x 12 km, (c) 20 km x 20 km, (d) 36 km x 36 km, (e) 48 km x 48 km, (f) 100 km x 100 km, (g) 200 km x 200 km, and (h) 300 km x 300 km. One-to-one line is shown in black.

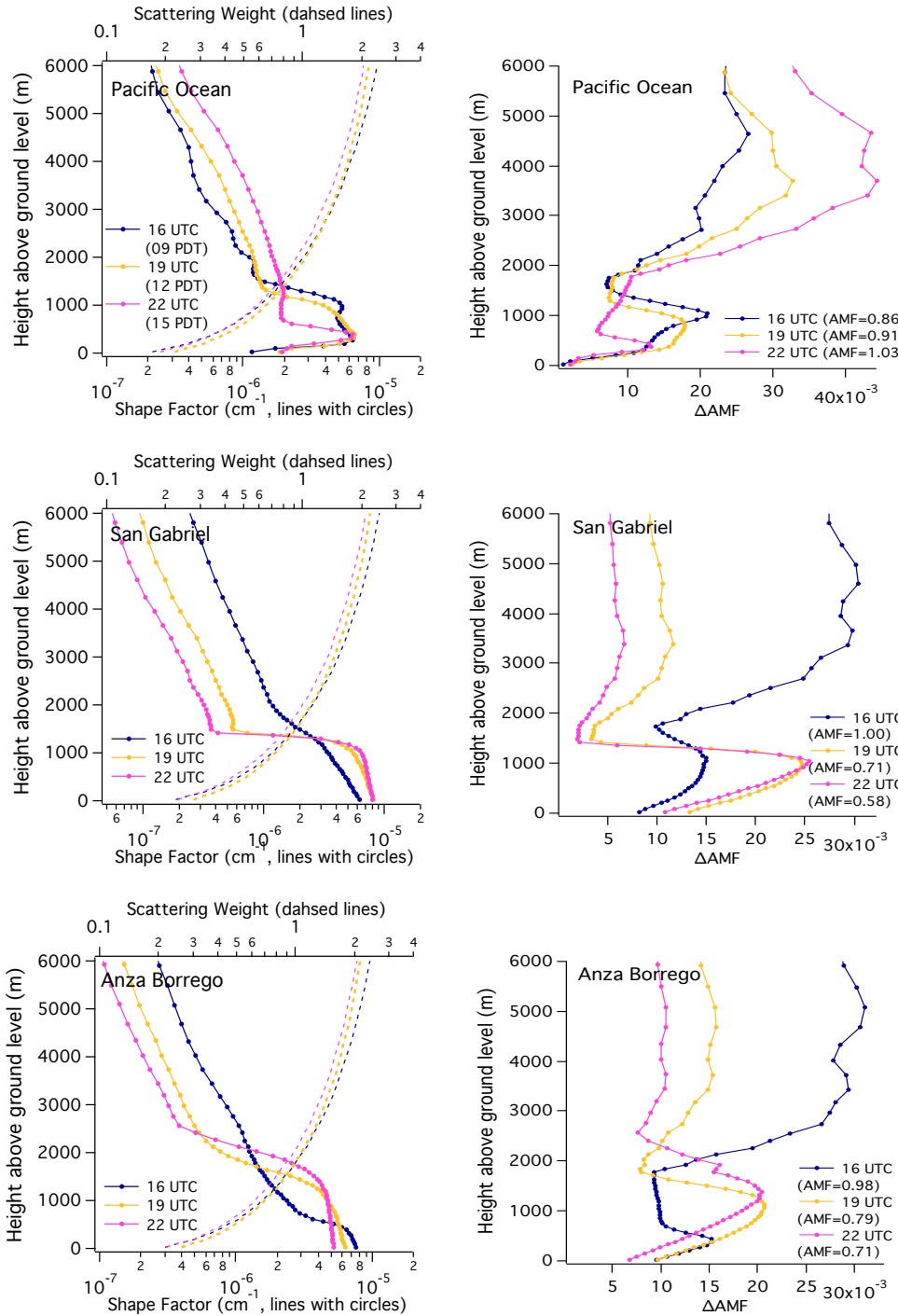

Figure 8. Vertical profiles of (left) shape factor and scattering weight and (right) $\Delta AMF_i$ (discrete increment of AMF) at North Pacific Ocean, San Gabriel Mountains, and Anza Borrego Desert State Park. Scattering weights multiplied by geometric AMF are shown. Dashed lines and solid lines with symbols in the left panel denote the scattering weight and shape factor, respectively. Total (or tropospheric) AMF values are shown in legends in the right panel.

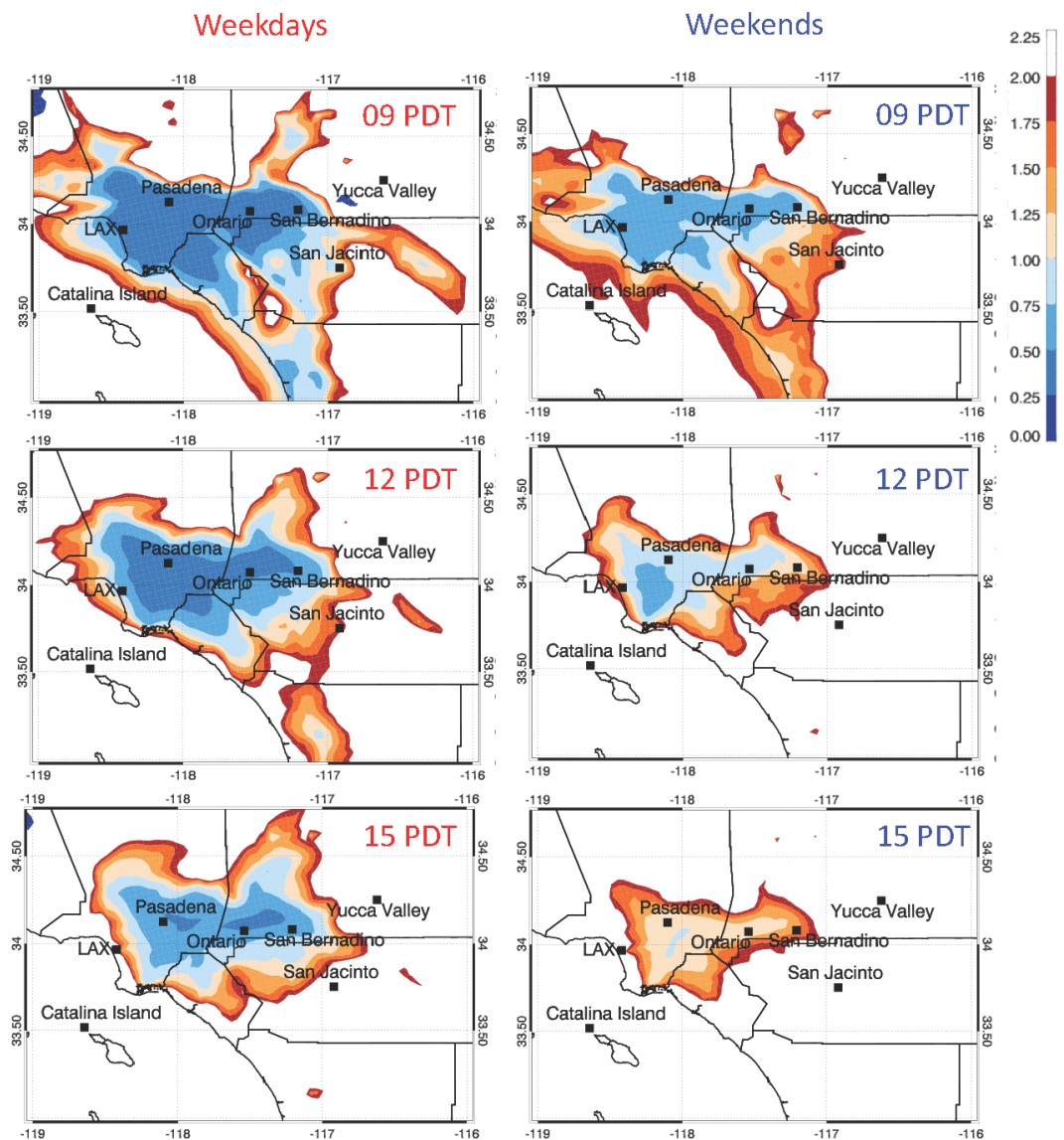

Figure 9. Spatial distributions of the ratios of the model HCHO column to $NO_2$ column during weekdays (left) and weekends (right) at 09 PDT, 12 PDT, and 15 PDT for May-June 2010. The light pink to red colored contours denote the area under the $NO_x$-limited chemical regime, while blue contours denote VOC-limited regions.

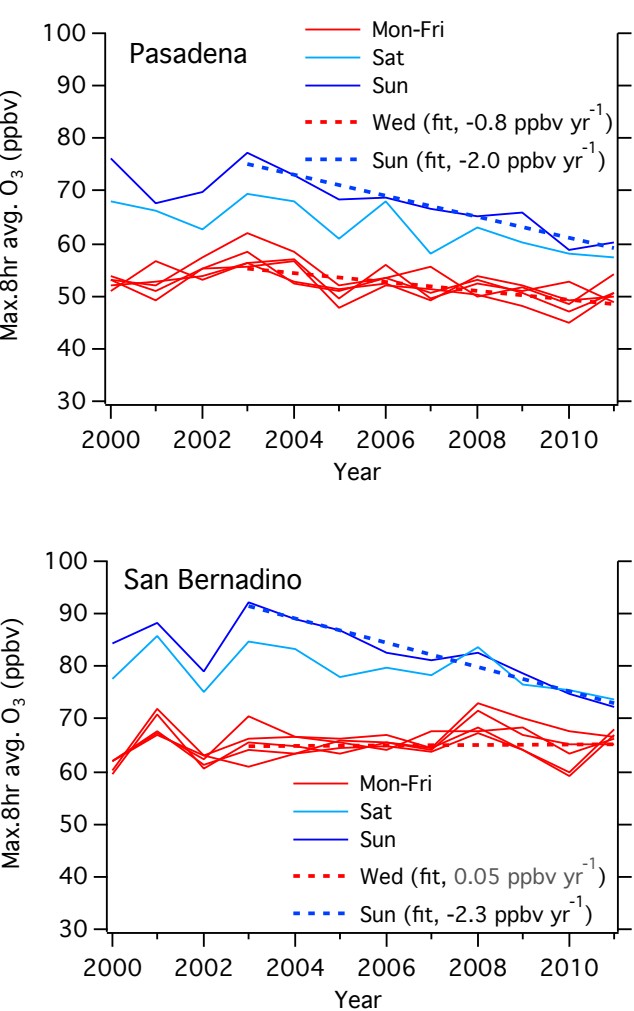

Figure 10. Decadal O₃ trends in Pasadena and San Bernardino during weekdays (red) and weekends (blue) are shown. The linear least square fits of O₃ for Wednesday and Sunday are plotted in dashed lines.

Table 1. Percentage (%) of intense HCHO plumes retained as the spatial resolution is changed from 4 km. Each column shows the fraction of the plumes retained at coarser resolutions. Here the plume is defined by the area in which the HCHO mixing ratio is greater than the reference HCHO volume mixing ratio (VMR) (1-6 ppb) at 4 km resolution. For example, the second column shows how much area at 8-200 km resolution has a HCHO VMR > 1 ppb when compared with the area with VMR > 1 ppb at 4 km resolution. Similarly, the last column shows how often a model HCHO VMR is greater than 6 ppb at 8-200 km resolution compared with the same plume of VMR > 6 ppb at 4 km resolution; all coarser resolutions (8-200 km) fail to capture this most intense plume. Only model HCHO results at 200 m above ground level at 19 UTC (12 PDT) are used. The areas with HCHO VMRs greater than 1, 2, 3, 4, 5, or 6 ppb are 92800, 29136, 12832, 4256, 848, or 64 km$^2$, respectively in the original simulations at 4 km resolution. The area of the domain is 143856 km$^2$.

| Spatial resolution (km) | Reference HCHO volume mixing ratio (ppb) at 4 km resolution | | | | | |
|---|---|---|---|---|---|---|
| | 1 | 2 | 3 | 4 | 5 | 6 |
| 8 | 98 (%) | 95 | 91 | 84 | 79 | 0 |
| 12 | 97 | 92 | 85 | 73 | 79 | 0 |
| 20 | 97 | 86 | 72 | 67 | 58 | 0 |
| 36 | 97 | 82 | 52 | 29 | 0 | 0 |
| 48 | 96 | 72 | 51 | 0 | 0 | 0 |
| 100 | 96 | 62 | 0 | 0 | 0 | 0 |
| 200 | 89 | 56 | 0 | 0 | 0 | 0 |
| 300 | 53 | 46 | 0 | 0 | 0 | 0 |

Table 2. Summary of air mass factors at 8 locations at 16-22 UTC (09-15 PDT). The results without/with aerosols impacts are also shown.

| Location | 16 UTC (09PDT) Aerosol | | 19 UTC (12 PDT) Aerosol | | 22 UTC (15 PDT) Aerosol | |
|---|---|---|---|---|---|---|
| | X | O | X | O | X | O |
| N. Pacific Ocean | 0.86 | 0.75 | 0.90 | 0.85 | 1.03 | 0.99 |
| Los Padres | 1.21 | 1.15 | 0.90 | 0.86 | 1.02 | 1.00 |
| Main St. | 0.70 | 0.60 | 0.61 | 0.54 | 0.69 | 0.62 |
| Pasadena | 0.71 | 0.62 | 0.60 | 0.53 | 0.66 | 0.58 |
| San Gabriel | 1.00 | 0.93 | 0.71 | 0.65 | 0.58 | 0.51 |
| San Bernardino | 1.07 | 1.02 | 0.89 | 0.86 | 0.69 | 0.66 |
| San Jacinto | 1.12 | 1.07 | 0.95 | 0.93 | 0.76 | 0.73 |
| Anza-Borrego | 0.98 | 0.91 | 0.79 | 0.75 | 0.71 | 0.66 |