# Peer review of "Impact of high-resolution *a priori* profiles on satellite-based formaldehyde retrievals"

_Atmospheric Chemistry and Physics, 2017_

## Referee Comment (RC1) · Anonymous Referee #2 · 26 Dec 2017

This manuscript by Kim and co-authors, discusses the impact of high-resolution *a priori* profiles on satellite formaldehyde retrievals. They use high resolution simulations from the Weather Research and Forecasting-Chemistry model (WRF-Chem) over California (4 km x 4km) during April and July 2010 to obtain the atmospheric information necessary to compute formaldehyde air mass factors (AMF). The paper includes a basic validation of the model using airborne PTR-MS (Proton-Transfer-Reaction Mass-Spectrometry) and long-path DOAS (LP-DOAS) measurements obtained during the California Nexus of Air Quality and Climate Change (CalNex) campaign. The paper continues discussing the AMF dependency with respect time of the day and the vertical distribution of formaldehyde as well as their spatial distribution. The paper finishes showing the dominant chemical regimes in the L.A. basin at different times of the day derived from WRF-Chem simulations as an example of the capabilities that future geostationary sensors will enable.

The paper is well written and tries to address an important question relevant for the future generation of air-quality satellite sensors such as the impact of high resolution a-priori information in the retrieval accuracy. However, some aspects of the paper should be improved before publication in ACP. These are summarized below with further discussion following for each section.

1. A better description of the radiative transfer calculations is needed. It is not clear how some of the most basic parameters needed for a radiative transfer calculation are treated, i.e. geometry and surface reflectance. Clarifications about how the wide spectral range is used is needed.
2. The discussion of WRF-Chem validation with CalNex data could be expanded with detailed description of the methodology used to match PTR-MS and LP-DOAS measurements with WRF-Chem simulations.
3. AMF calculations at 4 km x 4km pixels are shown but these are not compared with calculations at coarser resolution. There is no analysis included about the error in AMFs due to the spatial resolution of a priori vertical profile information. It will good to include such analysis. Furthermore, AMF calculations are affected by other sources of error such as surface reflectance or topography. This should be at least discussed in the text. Some conclusions and suggestions are qualitative and vague and should be backed up by further quantitative analysis.
4. Section 3.3 doesn't seams to belong to this paper. While it is important to highlight the capabilities of future satellite sensors it is not clear how that example provides any further information about the impact of high-resolution a priori profiles in satellite retrievals.

**Abstract:** With the evidence provided in the text the following sentence is not fully supported "Our analyses suggest that an air mass factor (AMF, a factor converting observed slant columns to vertical columns) based on fine spatial and temporal resolution *a priori* profiles can better capture the spatial distributions of the enhanced HCHO plumes in an urban area than the nearly

constant AMFs used for current operational products". High resolution AMFs are not compared with low resolution AMFs.

**Section 2.3:**

- At the wavelengths of interest for UV retrievals the surface and atmospheric thermal emission is not relevant. Why are they included in the simulations?
- "We adopt the spectral resolution of 0.2 nm and a spectral range of 300.5 – 365.5 nm". Typical formaldehyde satellite retrievals perform AMF calculations at one wavelength ~340nm. How are the calculations between 300.5 and 365.5 nm used? What is the impact of the 0.2 nm resolution? With typical fitting windows between ~328 nm to ~360 nm why is the ~300 nm to ~328 nm spectral range included?
- For each pixel what is the viewing geometry used? Is it assumed the longitude of a geostationary orbit to work out solar, viewing and azimuth angles? This is important information that needs to be included in the description. The similar scattering weights in figure 7 indicate small variations in the viewing geometries (solar angle).
- How is the surface reflectance modelled in the radiative transfer calculations? Is it assumed to be a Lambertian surface with wavelength dependency and time of the day dependency, is it assumed to be a BRDF?

**Section 3.1:**

- The description about how WRF-Chem and LP-DOAS measurements are collocated and compared should be expanded. There are at least three dimensions that should be considered: horizontal, vertical and temporal. Is the horizontal and vertical sampling of the LP-DOAS measurements accounted for? If so, how? Is there any filtering of LP-DOAS? How is the averaging in the time-domain done?
- Likewise for the comparison between WRF-Chem and aircraft data. There is no description about how WRF-Chem simulations and aircraft profiles are matched. It needs to be included to understand the significance of figure 2.

**Section 3.2:** As mentioned above these section should include an estimate of the AMF calculations sensitivity with respect to vertical profiles spatial resolution by discussing "high" and "low" spatial resolution cases.

- Page 13 line 7: "General features of the AMF distribution in the area do not change significantly when a constant surface pressure is used in the RT simulations (see Supplementary Material Figure S1)." This statement is qualitative. Can it be quantified? How is the vertical distribution of HCHO and other trace gases treated when using a constant surface pressure? Are total columns kept constant? What is the value of that surface pressure? Figure S1 says "Low Spectral Resolution". Nowhere in the text it is introduced a "Low Spectral Resolution" or "High Spectral Resolution" calculation.

- Page 14 line 17: "The AMF over the ocean increases with time from 0.86 at 09 PDT to 1.03 at 15 PDT as the HCHO mixing ratio decreases with time, probably due to transport of the plume from the ocean to the inland area." Could be discussed the effect on AMF calculation of the development of the marine boundary layer? Would it be possible to quantify transport using WRF-Chem to support this statement?
- Figures 4 and 5: While mixing ratios are interesting, the actual quantity considered in the AMF calculations is the number density. Could that be shown instead?
- Page 15, line 18: "These findings highlight the importance of using time-varying, high spatial resolution a priori profile information for the accurate retrieval of geostationary HCHO measurements." While there is some quantitative analysis of the importance of using time-varying profiles by showing calculations at 3 different times, there is not such analysis for different spatial resolutions.
- Page 16, line 8: "The dependence of the AMF value on the profile shape is similar at each time of day." Would it be possible to provide a quantitative analysis backing it up?
- Page 16, line 13: "For UV-VIS retrievals, it is generally assumed that only the vertical profile shape, rather than the absolute magnitude of the absorber, affects the value of the AMF." UV-VIS retrievals, as shown in equations 1 and 2, consider the absolute magnitude of the absorber $\Omega_v$. It is true that for similar shapes of the vertical distribution of number densities of HCHO columns the values of $S_z(z)$ will remain constant since it is a normalized quantity. However, a consequence of the atmospheric chemistry, sources and sinks of HCHO is that high total columns and low total columns are generally linked to different shape factors.
- Page 20, line 20: "It is likely that the actual impact of aerosols on the AMF is relatively small when compared with other factors examined here." This is a qualitative statement that should be backed up with data. Otherwise it should be removed. Kwon et al., 2017 showed the impact of aerosols over East Asia not to be negligible changing columns up to 47%.

**Section 3.3:**

- Page 21, line 21: "Figure 9 shows 2000-2010 trends in surface $O_3$ from monitors in Pasadena and San Bernardino." A brief description of those monitors and their datasets should be added.

Technical comments:

Page 3, Line 2: remove the before sources.

Page 3, Line 5: Add reference for EPA HAP

Page 3, Line 8: Add reference with HCHO atmospheric chemistry.

Page 3, Line 12: Add reference to X. Jin et al., 2017 doi:10.1002/2017JD026720

Page 4, Line 10: Add reference to A. Lorente et al., 2017 doi:10.5194/amt-10-759-2017

Page 5, Line 10: Add reference for TROPOMI.

Page 21, Line 11: Add reference to X. Jin et al., as above

Page 24, Line 2: The "authors think" should be the "authors thank".

Page 25, Line 25: The year of Borbon et al., should be 2013.

Page 37, Figure 3: It will be good to include the corresponding PDT values as well.

Page 41, Figure 7: Where it says slopre factor it should say shape factor.

Line 68, please include reference to Razavi et al., 2011 (first HCOOH retrievals from IASI).

Line 71, please include Gonzalez Abad et al., 2009 in ACE-FTS papers.

Line 98, please include citation about IASI $CO_2$ retrievals.

Line 118, correct typo (Pommier et al., 2016).

Line 141, actives to become active.

Line 206, should read "Both biases are however" instead of "Both biases is howeve"

Line 282, please specify which other studies.

Figure 2, include units in plots.

Figure 4, please include units in plots.

---

## Referee Comment (RC2) · Anonymous Referee #1 · 9 Jan 2018

The comment was uploaded in the form of a supplement:
https://www.atmos-chem-phys-discuss.net/acp-2017-982/acp-2017-982-RC2-
supplement.pdf

---

## Author Comment (AC1) · 5 Apr 2018

We thank the reviewers for the comments that greatly improved the manuscript. Our responses to the reviewer's comments below are highlighted in blue.

1. *A better description of the radiative transfer calculations is needed. It is not clear how some of the most basic parameters needed for a radiative transfer calculation are treated, i.e. geometry and surface reflectance. Clarifications about how the wide spectral range is used is needed.*

   → The missing information is added in the revised manuscript. Solar zenith angles are 52.8°, 16.7°, and 28.8° at 16, 19, 22 UTC, respectively. Relative azimuth angles are 56.6°, 15.5°, 246.1° at 16, 19, 22 UTC, respectively. Viewing zenith angle in VLIDORT is 46.5°. We assume a constant surface reflectance of 0.05 across the domain. The AMF presented in the manuscript is selected at 340 nm similar to the current satellite retrieval. This information is included (Page 11, Line 9 – 19) in the revised manuscript.

2. *The discussion of WRF-Chem validation with CalNex data could be expanded with the detailed description of the methodology used to match PTR-MS and LP-DOAS measurements with WRF-Chem simulations.*

   → More detailed explanations of how the model results are compared with the PTR-MS and LP-DOAS are added. The model results are sampled at the times and locations nearest the observations. The PTR-MS measurement data onboard the P3 aircraft and the sampled model data are averaged at the model spatial resolution (horizontal and vertical) to allow one-to-one comparison of the observations and model results. The LP-DOAS data have been averaged over the upper light path from 35 m AGL (Millikan Library at Caltech) to 225 m AGL (water tank in Altadena) and have been averaged for one hour prior to the comparison with the model results. The model values on the vertical levels corresponding to 35 m to 225 m AGL are averaged for comparison with the LP-DOAS data. The model value from the 4 km x 4 km horizontal grid cell containing Millikan Library at Caltech is selected for the comparison with the LP-DOAS observations. This information is now included in the revised manuscript (P 7, L 14-18 and P 8, L 13-19).

3. *AMF calculations at 4km x 4km pixels are shown but these are not compared with calculations at coarser resolution. There is no analysis included about the error in AMFs due to the spatial resolution of a priori vertical profile information. It will be good to include such analysis. Furthermore, AMF calculations are affected by other sources of error such as surface reflectance or topography. This should be at least discussed in the text. Some conclusions and suggestions are qualitative and vague and should be backed up by further quantitative analysis.*

→ Operational HCHO retrievals use global model simulations at roughly 1°-3° grid size as *a priori* profiles, which are ~1000 times larger than the spatial resolution in our study (4 km x 4 km). Thus, we include "fine resolution" in the title. Following reviewer's comments, we added more discussion of this spatial resolution effect in the revised manuscript.

To understand the effect of spatial resolution, we compare the AMF from global model results (at 2° latitude x 2.5° longitude resolution) used as the *a priori* in the Smithsonian Astrophysical Observatory (SAO) OMI formaldehyde retrieval (Gonzalez Abad et al., 2015) with the AMF from this study in the LA Basin. In contrast to the AMF in this study (see Figure 4 in the manuscript), the AMF in the SAO OMI formaldehyde retrieval does not vary much across the Basin and is close to 1 (see Figure R1 below, which is Figure S3 in the Supporting Material). The average AMF from the OMI SAO product for the domain (33.5N-34.5N, 117W-118.5W) is 1.12, while the same domain average AMF from this study is 0.76. Using the AMF in this study, the domain average HCHO column increases by 47%, and up to ~100% at finer scales, compared with the SAO OMI HCHO column. The vertical HCHO profile in the OMI SAO product is almost constant across the domain, while the model profile at 4 km x 4 km resolution varies substantially. This discussion is included in the revised manuscript (P15, L9-P16, L4).

We discuss the spatial resolution effect on the intensity of HCHO plumes quantitatively as suggested by both reviewers. Figure R2 below (Figure 7 in the revised manuscript) demonstrates a scatter of HCHO mixing ratios at 4 km x 4 km resolution on increasingly coarser grid resolutions from 8 km to 300 km. Here the values for these coarser grids are generated from the spatial averages of the original model results at 4 km resolution in this study. The scatter of mixing ratios increases noticeably at grid resolutions ≥ 20 km. For

example, the mixing ratios at 4 km resolution vary from 1 to 6 ppb while those at 100 km resolution are between 0 and 3 ppb.

Table R1 (Table 1 in the revised manuscript) summarizes the efficiency of capturing the plumes that have HCHO volume mixing ratios (VMRs) greater than the reference values at each spatial grid resolution. Of particular importance are the reference values of 2 ppb and greater for which the AMF is greatly reduced. Table 1 indicates that a grid size ≤ 12 km can capture plumes of HCHO with VMRs > 4 ppb or 5 ppb at 4 km with an efficiency of more than 70%. If the grid size is 8 km, plumes of 1-5 ppb are detected with an efficiency of ~80%. If the grid size is greater than 100 km, it does not capture plumes with VMR > 2 ppb at this urban location. Thus, the AMF using coarse resolutions ≥ 100 km is about 1 because of the low HCHO VMR < 2 ppb.

Currently the typical spatial resolution of regional-scale models for the viewing domain of geostationary satellites like TEMPO (e.g., air quality forecast models for the U.S.) is 12-30 km in latitude and longitude. Our recommendation is to select the finest resolution available, and ideally 4 km. Model simulations at 4 km resolution are computationally expensive for a geostationary satellite's viewing domain and high quality model input data may not be readily available at this resolution (e.g., the emission inventory). At a minimum, model simulations at 8-12 km resolution should be tested for their ability to provide *a priori* profiles for next generation environmental geostationary satellite retrievals if computing resources are available.

The above text is included in the revised manuscript (P19, L12- P20, L16).

[Figure]

Figure R1. Comparison of the AMF in the OMI operational product (filled square at the center of the OMI swath) with the AMF from this study. An OMI pixel is 24 km x 13 km at nadir and the pixel size increases on either side of this point. The OMI AMF is about 1 on average (blue colors in the color scale used here).

[Figure]

Figure R2. Comparison of HCHO mixing ratios at 4 km x 4 km resolution with mixing ratios at coarser resolutions of (a) 8 km x 8 km, (b) 12 km x 12 km, (c) 20 km x 20 km, (d) 36 km x 36 km, (e) 48 km x 48 km, (f) 100 km x 100 km, (g) 200 km x 200 km, and (h) 300 km x 300 km. The one-to-one line is shown in black.

Table R1. Percentage (%) of intense HCHO plumes retained as the spatial resolution is changed from 4 km. Each column shows the fraction of the plumes retained at coarser resolutions. Here the plume is defined by the area in which the HCHO mixing ratio is greater than the reference HCHO volume mixing ratio (VMR) (1-6 ppb) at 4 km resolution. For example, the second column shows how much area at 8-200 km resolution has a HCHO VMR > 1 ppb when compared with the area with VMR > 1 ppb at 4 km resolution. Similarly, the last column shows how often a model HCHO VMR is greater than 6 ppb at 8-200 km resolution compared with the same plume of VMR > 6 ppb at 4 km resolution; all coarser resolutions (8-200 km) fail to capture this most intense plume. Only model HCHO results at 200 m above ground level at 19 UTC (12 PDT) are used. The areas with HCHO VMRs greater than 1, 2, 3, 4, 5, or 6 ppb are 92800, 29136, 12832, 4256, 848, or 64 $km^2$, respectively in the original simulations at 4 km resolution. The area of the domain is 143856 $km^2$.

| Spatial resolution (km) | Reference HCHO volume mixing ratio (ppb) at 4 km resolution | | | | | |
|---|---|---|---|---|---|---|
| | 1 | 2 | 3 | 4 | 5 | 6 |
| 8 | 98 (%) | 95 | 91 | 84 | 79 | 0 |
| 12 | 97 | 92 | 85 | 73 | 79 | 0 |
| 20 | 97 | 86 | 72 | 67 | 58 | 0 |
| 36 | 97 | 82 | 52 | 29 | 0 | 0 |
| 48 | 96 | 72 | 51 | 0 | 0 | 0 |
| 100 | 96 | 62 | 0 | 0 | 0 | 0 |
| 200 | 89 | 56 | 0 | 0 | 0 | 0 |
| 300 | 53 | 46 | 0 | 0 | 0 | 0 |

4. *Section 3.3 doesn't seem to belong to this paper. While it is important to highlight the capabilities of future satellite sensors it is not clear how that example provides any further information about the impact of high-resolution a priori profiles in satellite retrievals.*

→ The point we are making in this section is that the ability to spatially resolve urban plumes with improved satellite retrievals using fine-resolution *a priori* profiles can provide information relevant to tropospheric ozone chemistry and environmental policy at an urban scale. For example, resolving fine-scale plume structures helps to understand the chemical regimes leading to surface ozone production across the LA basin. We therefore have decided to retain this section in the revised manuscript.

**Abstract:** With the evidence provided in the text the following sentence is not fully supported "Our analyses suggest that an air mass factor (AMF, a factor converting observed slant columns to vertical columns) based on fine spatial and temporal resolution *a priori* profiles can better capture the spatial distributions of the enhanced HCHO plumes in an urban area than the nearly constant AMFs used for current operational products". High resolution AMFs are not compared with low resolution AMFs.

→ See our response above. We now compare our high resolution AMF with the lower resolution AMF used in the SAO OMI HCHO product. In addition, the effect of spatial resolution on the ability to capture the intensity of HCHO plumes is also included in the revised manuscript.

**Section 2.3:**

• At the wavelengths of interest for UV retrievals the surface and atmospheric thermal emission is not relevant. Why are they included in the simulations?

→ The reviewer is correct. We did not include thermal emission. We omitted this sentence in the revised manuscript.

• "We adopt the spectral resolution of 0.2 nm and a spectral range of 300.5 – 365.5 nm".

Typical formaldehyde satellite retrievals perform AMF calculations at one wavelength ~340nm.

How are the calculations between 300.5 and 365.5 nm used? What is the impact of the 0.2 nm resolution? With typical fitting windows between ~328 nm to ~360 nm why is the ~300 nm to ~328 nm spectral range included?

→ Our calculations are simply done for a spectral range covering wide enough to cover the typical fitting window. We compared the AMF values at several wavelengths and found them to be similar, so we present the AMF at 340 nm in the manuscript.

We initially used a spectral resolution of 0.05 nm. To reduce the computation time, the spectral resolution was reduced from 0.05 nm to 0.2 nm. The spectral resolution did not affect the AMF values we derived in this study.

In the revised manuscript, we clarify the wavelength at which the HCHO AMF is selected and we omit unnecessary notations of "low" and "high" spectral resolution in the plots.

• For each pixel what is the viewing geometry used? Is it assumed the longitude of a geostationary orbit to work out solar, viewing and azimuth angles? This is important information that needs to be included in the description. The similar scattering weights in figure 7 indicate small variations in the viewing geometries (solar angle).

→ As mentioned above, solar zenith angles are 52.8°, 16.7°, and 28.8° at 16, 19, 22 UTC, respectively. Relative azimuth angles are 56.6°, 15.5°, 246.1° at 16, 19, 22 UTC, respectively. Viewing zenith angle in the VLIDORT model is a constant 46.5°. We now specify the information about viewing geometry in the manuscript.

• How is the surface reflectance modelled in the radiative transfer calculations? Is it assumed to be a Lambertian surface with wavelength dependency and time of the day dependency, is it assumed to be a BRDF?

→ To focus on the effect of profile shape, we kept the surface reflectivity constant at 0.05 across the domain. The following text is included in the revised manuscript:
We assume a constant surface reflectance of 0.05 across the domain. For snow-covered mountain top and desert areas, the surface reflectivity can be larger than 0.05, which would increase the sensitivity of satellite HCHO observations to the surface, and in turn would increase the AMF and further modify the spatial distribution of AMF in Southern California. The sensitivity of the

HCHO AMF to the surface reflectivity for this area needs to be pursued in future study using data adequate for the TEMPO HCHO retrieval (P11, L9- P11, L19).

**Section 3.1:**

• The description about how WRF-Chem and LP-DOAS measurements are collocated and compared should be expanded. There are at least three dimensions that should be considered: horizontal, vertical and temporal. Is the horizontal and vertical sampling of the LP-DOAS measurements accounted for? If so, how? Is there any filtering of LP-DOAS? How is the averaging in the time-domain done?

→ The description of these comparisons is now included in the revised manuscript. See the responses above.

• Likewise for the comparison between WRF-Chem and aircraft data. There is no description about how WRF-Chem simulations and aircraft profiles are matched. It needs to be included to understand the significance of figure 2.

→ The description of these comparisons is now included in the revised manuscript. See the responses above.

**Section 3.2:** As mentioned above these section should include an estimate of the AMF calculations sensitivity with respect to vertical profiles spatial resolution by discussing "high" and "low" spatial resolution cases.

→ In the revised manuscript and responses above, we added a discussion of the effects of varying spatial resolution by comparing with the OMI operational product and by analyzing the sensitivity of HCHO plume detection to the spatial resolution of the model.

• Page 13 line 7: "General features of the AMF distribution in the area do not change significantly when a constant surface pressure is used in the RT simulations (see Supplementary Material Figure S1)." This statement is qualitative. Can it be quantified? How is the vertical distribution of HCHO and other trace gases treated when using a constant surface pressure? Are total columns kept constant? What is the value of that surface pressure? Figure S1 says

"Low Spectral Resolution". Nowhere in the text it is introduced a "Low Spectral Resolution" or "High Spectral Resolution" calculation.

→ To test the effect of surface pressure, we switched vertical profiles of pressure, temperature, and height in all grid cells with those at one oceanic location (32°N, 120°W) in the input files to VLIDORT. The constant surface pressure value is 1016 hPa. The quantitative analysis of the effect of a constant surface pressure is now included in Figure R3 (Figure S2 in Supporting Material). The differences between the AMF with constant surface pressure and the original AMF are generally less than 10%. 82% (99%) of the domain has AMF differences of less than 5% (10%).

We also added a discussion and quantitative analysis of the impact of the bottom-up emission inventory in the revised manuscript. The spatial pattern of AMF was not strongly affected by the currently available bottom-up emission inventory used to generate the WRF-Chem HCHO profiles in our study (see Supplementary Material Figure S1 and S2). 95% (98%) of the area shows differences in AMF of less than 5% (10%). The impact of the bottom-up emission inventory was larger in Barkley et al. (2012), who compared the effect of using various isoprene emission inventories over tropical South America for satellite HCHO retrievals. In general, Barkley et al. (2012) found an average difference in the HCHO columns of ±20% and up to 45% in individual locations. The role that the bottom-up emission inventory plays in the AMF calculation therefore depends on the quality (accuracy) of the emission inventories and their impacts on the profile shapes.

Regarding the spectral resolution of VLIDORT, high (low) resolution is 0.05 nm (0.2 nm). For our AMF calculations, this resolution impact is trivial. Following the reviewer's comment, we omitted the "Low" and "High" portions in the manuscript.

[Figure]

Figure R3. Histogram of (left) differences between the default AMF and the AMF derived using constant surface pressure, and (right) differences between the default AMF and the AMF derived using the NEI11 inventory (with lower VOC emissions than our default inventory) at 19 UTC (12 PDT).

• Page 14 line 17: "The AMF over the ocean increases with time from 0.86 at 09 PDT to 1.03 at 15 PDT as the HCHO mixing ratio decreases with time, probably due to transport of the plume from the ocean to the inland area." Could be discussed the effect on AMF calculation of the development of the marine boundary layer? Would it be possible to quantify transport using WRF-Chem to support this statement?

→ In response to the reviewer's suggestion, we analyzed the development of the marine boundary layer and transport. Figure R4 (Figure S5 in Supplementary Material) shows that HCHO mixing ratios above 200 m altitude decrease with time from 06 PDT to 16 PDT. The thermal structure, as shown in the vertical profiles of potential temperature, does not vary much with time. But wind speed changes substantially throughout the day. In the morning, the peak wind speed occurs at ~1.5 km, but the highest wind speeds move lower in altitude (< 500 m) in the afternoon. In the lower atmosphere (altitude < 500 m), wind speed increases during the day from 6 m/s to 15 m/s and wind direction changes from northerly to northwesterly during the same time period. These strong wind changes throughout the day enhance transport of HCHO,

while chemical formation is not high enough to compensate the wind-driven loss of HCHO in this area. This discussion is included in the revised manuscript.

[Figure]

Figure R4. Diurnal variations (06 PDT to 16 PDT) of vertical profiles of HCHO mixing ratio, potential temperature, wind speed, and wind direction over the North Pacific Ocean region.

• Figures 4 and 5: While mixing ratios are interesting, the actual quantity considered in the AMF calculations is the number density. Could that be shown instead?

→ We included the plots in terms of number density below as Figure R5-R7 and in the Supplementary Material (Figure S4, S6, S7). Mixing ratios of HCHO are also widely used. Therefore, we continue to use mixing ratio in the figures of the main text and provide the plots in terms of number density in the Supplementary Material.

[Figure]

Figure R5. Vertical profiles of HCHO number density are shown for various point of interest, similar to Figure 4 in the main manuscript.

[Figure]

Figure R6. Vertical profiles of HCHO number density averaged for the AMF value intervals (shown in the legends) at 16, 19, and 22 UTC (left to right) as a function of altitude above ground level. Thick lines with symbols are averages and thin dotted lines are one standard deviations. This figure is similar to Figure 5 in the main manuscript except that HCHO number density is shown instead of mixing ratio.

[Figure]

Figure R7. The relationship between the HCHO AMF and model HCHO volume mixing ratio at ~ 200 m altitude. Different colors denote different times. This figure is similar to Figure 6 in the main manuscript except that HCHO number density is shown instead of HCHO mixing ratio.

• Page 15, line 18: "These findings highlight the importance of using time-varying, high spatial resolution a priori profile information for the accurate retrieval of geostationary HCHO measurements." While there is some quantitative analysis of the importance of using time-varying profiles by showing calculations at 3 different times, there is not such analysis for different spatial resolutions.

→ Discussion of the spatial resolution effect is now included. See our responses above.

• Page 16, line 8: "The dependence of the AMF value on the profile shape is similar at each time of day." Would it be possible to provide a quantitative analysis backing it up?

→ We modified this section to make the meaning clearer, as follows:

The dependence of the AMF value on the profile shape is similar at each time of day: the higher AMF is related to lower HCHO mixing ratios (or number densities) in the atmospheric boundary layer (up to 1-3 km altitude AGL). More quantitative analysis is shown below.

Using all available data points, we investigate the relationship between AMF and the HCHO mixing ratio at 200 m in the boundary layer at different times of day in Figure 6 [see Figure S7 in Supporting Material for similar plots in terms of number density (molecules cm$^{-3}$)]. Figure 6 illustrates that as the HCHO mixing ratio increases, the AMF decreases. At all times investigated, AMF is anti-correlated with HCHO mixing ratio (or number density). Correlation coefficients between AMF and HCHO mixing ratio are -0.68, -0.85 and -0.84 at 16 (09), 19 (12), and 22 (15) UTC (PDT).

• Page 16, line 13: "For UV-VIS retrievals, it is generally assumed that only the vertical profile shape, rather than the absolute magnitude of the absorber, affects the value of the AMF." UV-VIS retrievals, as shown in equations 1 and 2, consider the absolute magnitude of the absorber $\Omega v$. It is true that for similar shapes of the vertical distribution of number densities of HCHO columns the values of $S_Z(z)$ will remain constant since it is a normalized quantity. However, a consequence of the atmospheric chemistry, sources and sinks of HCHO is that high total columns and low total columns are generally linked to different shape factors.

→ We agree with the reviewer. The absolute value of HCHO columns (or HCHO concentrations in the boundary layer) is related to the shape factor. Ironically, in general, the accuracy of a priori profile (absolute value) is rather neglected and is not analyzed. Since the original sentence can be misinterpreted, we modified it in the revised manuscript as follows:

For UV-VIS retrievals, it is well known that the vertical profile shape affects the value of the AMF. Our study suggests a strong anti-correlation between the absolute concentration and the AMF: the AMF is low in the area of intense HCHO plumes. The changes in the absolute HCHO concentrations in the boundary layer (altitude AGL < 1-3 km) strongly modify profile shapes, which in turn affect AMF substantially.

• Page 20, line 20: "It is likely that the actual impact of aerosols on the AMF is relatively small when compared with other factors examined here." This is a qualitative statement that should be backed up with data. Otherwise it should be removed. Kwon et al., 2017 showed the impact of aerosols over East Asia not to be negligible changing columns up to 47%.

→ This statement is supported with Table R2 below (Table 2 in the manuscript), the plots below (Figure R8, also Figure S8 in the Supplementary Material) and additional discussion in the manuscript. We now mention that the impact of aerosols can be large over East Asia and refer to Kwon et al. (2017). The text included in the revised manuscript is as follows:

Although the focus of this manuscript is on the shape factor, we also investigate the impact of aerosol loading on AMF for the 8 sites shown in Figure 4. When the aerosol optical properties from the model results are incorporated in our RT model calculations, the AMF is reduced by ~10% at the N. Main St. and Pasadena sites and by < 10% at other sites (Table 2). The aerosol optical depth, single scattering albedo, and asymmetry factor calculated from the model results for the 8 sites are about 0.5, 0.9, and 0.7, respectively. These are close to the values suggested as the most probable atmospheric conditions in the LA Basin (see Table 4 in Baidar et al., 2013). Because the model aerosol results were not thoroughly evaluated and optimized and only 8 sites were tested, the analysis of aerosol impact in this study is limited. It is possible that some of the simulated aerosol components are overestimated, because the emission inventory is not fully up to date for primary aerosol emissions and aerosol precursor gases (e.g., overestimations of black carbon and $SO_2$ by a least a factor of 3). Meanwhile, the AMF changes from the values at 16 UTC (09PT) due to diurnal variations in *a priori* profile shape range from -40% to 20% (Table 2). It is likely that the impact of aerosols on the AMF is relatively small when compared with the impact of the profile shape factor examined in this study for the LA basin. De Smedt et al. (2015) and Wang et al. (2017) also reported the importance of *a piori* profile shapes for an improvement of satellite-based HCHO retrievals in Beijing, Xianghe, Wuxi in China. Kwon et al. (2017) demonstrated that the impact of aerosol loading on HCHO AMF can be large over East Asia in contrast to our study for the LA basin.

Table R2. Summary of air mass factors at 8 locations at 16-22 UTC (09-15 PDT). The results without/with aerosols impacts are also shown.

| Location | 16 UTC (09PDT) | | 19 UTC (12 PDT) | | 22 UTC (15 PDT) | |
| --- | --- | --- | --- | --- | --- | --- |
| | Aerosol | | Aerosol | | Aerosol | |
| | X | O | X | O | X | O |
| N. Pacific Ocean | 0.86 | 0.75 | 0.90 | 0.85 | 1.03 | 0.99 |
| Los Padres | 1.21 | 1.15 | 0.90 | 0.86 | 1.02 | 1.00 |
| Main St. | 0.70 | 0.60 | 0.61 | 0.54 | 0.69 | 0.62 |
| Pasadena | 0.71 | 0.62 | 0.60 | 0.53 | 0.66 | 0.58 |
| San Gabriel | 1.00 | 0.93 | 0.71 | 0.65 | 0.58 | 0.51 |
| San Bernardino | 1.07 | 1.02 | 0.89 | 0.86 | 0.69 | 0.66 |
| San Jacinto | 1.12 | 1.07 | 0.95 | 0.93 | 0.76 | 0.73 |
| Anza-Borrego | 0.98 | 0.91 | 0.79 | 0.75 | 0.71 | 0.66 |

[Figure]

Figure R8. (Top) AMF at 8 sites in the domain at 9, 12, and 15 PDT without/with aerosol impacts. Filled (open) square denote AMF with (without) aerosol impacts. (Bottom) changes in AMF (%) with time. Black (red) open square denotes changes of AMF between 9 and 12 PDT (15PDT).

**Section 3.3:**

• Page 21, line 21: "Figure 9 shows 2000-2010 trends in surface O3 from monitors in Pasadena and San Bernardino." A brief description of those monitors and their datasets should be added.

→ We added the information in the revised manuscript. The hourly $O_3$ data from the South Coast Air Quality Management District (AQMD) monitoring network (http://www.arb.ca.gov/aqmis2/aqdselect.php) are utilized for the trend study. Details on standard procedures for maintaining and operating air monitoring stations and specific instrumentations are provided in the CARB air monitoring web manual (http://www.arb.ca.gov/airwebmanual/index.php). The locations of the sites and the data are shown in Auxiliary Material in Kim et al. (2016).

Technical comments:

Page 3, Line 2: remove the before sources.

→ "the" is removed.

Page 3, Line 5: Add reference for EPA HAP

→ A reference is added.

Technical Support Document EPA's 2011 National-scale Air Toxics Assessment, 2011 NATA TSD; United States Environmental Protection Agency: United States, 2015; https://www.epa.gov/sites/production/ files/2015-12/documents/2011-nata-tsd.pdf

Page 3, Line 8: Add reference with HCHO atmospheric chemistry.

→ A reference is added.

Wolfe, G. M., Kaiser, J., Hanisco, T. F., Keutsch, F. N., de Gouw, J. A., Gilman, J. B., Graus, M., Hatch, C. D., Holloway, J., Horowitz, L. W., Lee, B. H., Lerner, B. M., Lopez-Hilifiker, F., Mao, J., Marvin, M. R., Peischl, J., Pollack, I. B., Roberts, J. M., Ryerson, T. B., Thornton, J. A., Veres, P. R., and Warneke, C.: Formaldehyde production from isoprene oxidation across $NO_x$ regimes, Atmos. Chem. Phys., 16, 2597-2610, https://doi.org/10.5194/acp-16-2597-2016, 2016.

Page 3, Line 12: Add reference to X. Jin et al., 2017 doi:10.1002/2017JD026720

→ Jin et al. (2017) is added.

Page 4, Line 10: Add reference to A. Lorente et al., 2017 doi:10.5194/amt-10-759-2017

→ Lorente et al. (2017) is added.

Page 5, Line 10: Add reference for TROPOMI.

→ A reference (Veefkind et al., 2012) is included in the text.

TROPOMI on the ESA Sentinel-5 Precursor: A GMES mission for global observations of the atmospheric composition for climate, air quality and ozone layer applications; Veefkind, J.P, et al. ; *Remote Sensing of Environment 120 (2012) 70-83*

Page 21, Line 11: Add reference to X. Jin et al., as above

→ Jin et al. (2017) is added.

Page 24, Line 2: The "authors think" should be the "authors thank".

→ Corrected.

Page 25, Line 25: The year of Borbon et al., should be 2013.

→ Corrected.

Page 37, Figure 3: It will be good to include the corresponding PDT values as well.

→ PDT values are added in the figure captions.

Page 41, Figure 7: Where it says slope factor it should say shape factor.

→ Corrected. It is Figure 8 in the revised manuscript.

The comments below do not seem to be relevant to our manuscript. Thus, we did not respond to these comments.

Line 68, please include reference to Razavi et al., 2011 (first HCOOH retrievals from IASI).
Line 71, please include Gonzalez Abad et al., 2009 in ACE-FTS papers.
Line 98, please include citation about IASI CO2 retrievals.
Line 118, correct typo (Pommier et al., 2016).

Line 141, actives to become active.
Line 206, should read "Both biases are however" instead of "Both biases is howeve"
Line 282, please specify which other studies.
Figure 2, include units in plots.
Figure 4, please include units in plots.

**Reference in the response and newly added in the revised manuscript**

Baidar, S., Oetjen, H., Coburn, S., Dix, B., Ortega, I., Sinreich, R., and Volkamer, R. (2013), The CU Airborne MAX-DOAS instrument: vertical profiling of aerosol extinction and trace gases, *Atmos. Meas. Tech.*, 6, 719-739, https://doi.org/10.5194/amt-6-719-2013.

Barkley, M. P., T. P. Kurosu, K. Chance, I. De Smedt, M. V. Roozendael, A. Arneth, D. Hagberg, and A. Guenther (2012), Assessing sources of uncertainty in formaldehyde air mass factors over tropical South America: Implications for top-down isoprene emission estimates, *J. Geophys. Res.-Atmos.*, 117, D13304, doi:10.1029/2011JD016827.

formaldehyde retrieval, *Atmos. Meas. Tech.*, 8, 19-32, https://doi.org/10.5194/amt-8-19-2015.

De Smedt, I., Stavrakou, T., Hendrick, F., Danckaert, T., Vlemmix, T., Pinardi, G., Theys, N., Lerot, C., Gielen, C., Vigouroux, C., Hermans, C., Fayt, C., Veefkind, P., Müller, J.-F., and Van Roozendael, M. (2015), Diurnal, seasonal and long-term variations of global formaldehyde columns inferred from combined OMI and GOME-2 observations, *Atmos. Chem. Phys.*, 15, 12519-12545, https://doi.org/10.5194/acp-15-12519-2015.

De Smedt, I., Theys, N., Yu, H., Danckaert, T., Lerot, C., Compernolle, S., Van Roozendael, M., Richter, A., Hilboll, A., Peters, E., Pedergnana, M., Loyola, D., Beirle, S., Wagner, T., Eskes, H., van Geffen, J., Boersma, K. F., and Veefkind, P. (2017), Algorithm Theoretical Baseline for formaldehyde retrievals from S5P TROPOMI and from the QA4ECV project, *Atmos. Meas. Tech. Discuss.*, https://doi.org/10.5194/amt-2017-393, in review.

González Abad, G., Liu, X., Chance, K., Wang, H., Kurosu, T. P., and Suleiman, R. (2015), Updated Smithsonian Astrophysical Observatory Ozone Monitoring Instrument (SAO OMI)

Jin, Xiaomeng, A. M. Fiore, L. T. Murray, L. C. Valin, L. N. Lamsal, B. Duncan, K. Folkert Boersma, I. De Smedt, G. Gonzalez Abad, K. Chance, and G. S. Tonnesen (2017), Evaluating a space-based indicator of surface ozone-NOx-VOC sensitivity over midlatitude source regions and application to decadal trends. *J. Geophys. Res.,* 122, 10,439-10,461. https://doi.org/10.1002/2017JD026720.

Lorente, A., Folkert Boersma, K., Yu, H., Dörner, S., Hilboll, A., Richter, A., Liu, M., Lamsal, L. N., Barkley, M., De Smedt, I., Van Roozendael, M., Wang, Y., Wagner, T., Beirle, S., Lin, J.-T., Krotkov, N., Stammes, P., Wang, P., Eskes, H. J., and Krol, M. (2017), Structural uncertainty in air mass factor calculation for NO$_2$ and HCHO satellite retrievals, *Atmos. Meas. Tech*., 10, 759-782, https://doi.org/10.5194/amt-10-759-2017.

US EPA (2015a), Technical Support Document EPA's 2011 National-scale Air Toxics Assessment, 2011 NATA TSD, https://www.epa.gov/sites/production/files/2015-12/documents/2011-nata-tsd.pdf.

Veefkind, J. P., et al. (2012), TROPOMI on the ESA Sentinel-5 Precursor: A GMES mission for global observations of the atmospheric composition for climate, air quality and ozone layer applications. *Remote Sensing of Environment 120, 70-83.*

Wang, Y., Beirle, S., Lampel, J., Koukouli, M., De Smedt, I., Theys, N., Li, A., Wu, D., Xie, P., Liu, C., Van Roozendael, M., Stavrakou, T., Müller, J.-F., and Wagner, T. (2017), Validation of OMI, GOME-2A and GOME-2B tropospheric NO$_2$, SO$_2$ and HCHO products using MAX-DOAS observations from 2011 to 2014 in Wuxi, China: investigation of the effects of priori profiles and aerosols on the satellite products, Atmos. Chem. Phys., 17, 5007-5033, https://doi.org/10.5194/acp-17-5007-2017.

Wolfe, G. M., Kaiser, J., Hanisco, T. F., Keutsch, F. N., de Gouw, J. A., Gilman, J. B., Graus, M., Hatch, C. D., Holloway, J., Horowitz, L. W., Lee, B. H., Lerner, B. M., Lopez-Hilifiker, F., Mao, J., Marvin, M. R., Peischl, J., Pollack, I. B., Roberts, J. M., Ryerson, T. B., Thornton, J. A., Veres, P. R., and Warneke, C. (2016), Formaldehyde production from isoprene oxidation across NO$_x$ regimes, Atmos. Chem. Phys., 16, 2597-2610, https://doi.org/10.5194/acp-16-2597-2016.

---

## Author Comment (AC2) · 5 Apr 2018

We thank the reviewers for the comments that greatly improved the manuscript. Our responses to the reviewer's comments below are highlighted in blue.

**General comments**

*The subject of the paper, studying the spatial and temporal variations of a priori HCHO profiles and their impact on AMF, is very relevant for current and future satellite retrievals. For their study, the authors used a regional model with a spatial resolution of 4x4km, at three different time of the day. The use of aircraft profiles and LP DOAS measurement to validate the model is giving to the paper an interesting added value to the paper, although their use is limited. However, while the title and the abstract promise to the reader for an evaluation of this resolution impact, the paper does not provide a quantitative answer. I would expect to get an estimate of the errors on AMF when the resolution is decreased in space or in time, with a distinction between both effects. What minimal model resolution is needed to capture the natural resolution of HCHO in the AMF (based on the model)?*

→ Both reviewers suggested to address the impact of spatial resolution in a quantitative manner. In the revised manuscript, we addressed this issue more in a systematic way. First, we compared the AMF from the SAO OMI HCHO retrieval (Gonzalez Abad et al., 2015) with the AMF in this study. In contrast to inhomogeneous AMF in this study, the AMF in the SAO OMI product does not vary much in the domain and is close to 1 (Figure R1 or Figure S3 in Supplementary Material). The average of AMF from the OMI SAO product for the domain (33.5N-34.5N, 117W-118.5W) is 1.12 while the same domain average of AMF from this study is 0.76. If AMF in this study is used, the HCHO column can increase by 47% on the domain-average (up to ~100% at a finer scale), compared with the OMI HCHO column. The vertical HCHO profile in the SAO OMI product is almost a constant in the domain while the model profile at 4 km x 4 km resolution varies substantially. This discussion is included in the revised manuscript P15, L9-P16, L4. As mentioned in the responses to the other reviewer's comments, the operational HCHO retrievals adopted global model results at roughly 1°-3° grid size as a priori profile, which are ~1000 times as large as the spatial resolution in our study (4 km x 4 km). Thus, we used "fine resolution" in the title. Second, we analyzed the effect of spatial resolution on capturing HCHO plumes

in the basin as the reviewer suggested. Figure 6 shows that AMF values are greatly reduced at HCHO mixing ratio of 2, 3 and 4 ppb. We examined the spatial resolutions at which the HCHO plumes of these critical levels of mixing ratio can be captured. The values for coarse grids (8 km – 300 km) are generated from the spatial averages of the original model results at 4 km resolution. Figure R2 and Table R1 (Figure 7 and Table 1 in the revised manuscript) indicate that the grid size ≤ 12 km can capture the plumes of HCHO VMR > 4 ppb or 5 ppb by more than 70%. If the grid size is 8 km, the plumes of 1-5 ppb are detected by ~80%. If the grid size is greater than 100 km, it does not capture the plume of VMR > 2 ppb at this urban location. Thus, the AMF using the coarse resolution ≥ 100 km is about 1 because of low concentration that is less than 2 ppb. Currently typical spatial resolution of regional-scale models for the viewing domain of the geostationary satellites (e.g., air quality forecast models for the U.S.) is 12-30 km in each latitude and longitude direction. Our recommendation is to select the resolution as close as 4 km. Since the model simulation at 4 km resolution is computationally expensive for the current geostationary satellite viewing domain and all of high quality input data to the model are not readily available at this resolution (e.g., emission inventory), the model simulations at 8-12 km resolution are recommended to test and improve the model simulations and finally acquire *a priori* profile for next generation environmental geostationary satellite retrievals if computing resources are available. This is included in the revised manuscript P19, L12- P20, L16.

[Figure]

Figure R1. Comparison of the AMF in the OMI operational product (filled square at the center of the OMI swath) with the AMF from this study. An OMI pixel is 24 km x 13 km at nadir and the pixel size increases on either side of this point. The OMI AMF is about 1 on average (blue colors in the color scale used here).

[Figure]

Figure R2. Comparison of HCHO mixing ratios at 4 km x 4 km resolution with mixing ratios at coarser resolutions of (a) 8 km x 8 km, (b) 12 km x 12 km, (c) 20 km x 20 km, (d) 36 km x 36 km, (e) 48 km x 48 km, (f) 100 km x 100 km, (g) 200 km x 200 km, and (h) 300 km x 300 km. The one-to-one line is shown in black.

Table R1. Percentage (%) of intense HCHO plumes retained as the spatial resolution is changed from 4 km. Each column shows the fraction of the plumes retained at coarser resolutions. Here the plume is defined by the area in which the HCHO mixing ratio is greater than the reference HCHO volume mixing ratio (VMR) (1-6 ppb) at 4 km resolution. For example, the second column shows how much area at 8-200 km resolution has a HCHO VMR > 1 ppb when compared with the area with VMR > 1 ppb at 4 km resolution. Similarly, the last column shows how often a model HCHO VMR is greater than 6 ppb at 8-200 km resolution compared with the same plume of VMR > 6 ppb at 4 km resolution; all coarser resolutions (8-200 km) fail to capture this most intense plume. Only model HCHO results at 200 m above ground level at 19 UTC (12 PDT) are used. The areas with HCHO VMRs greater than 1, 2, 3, 4, 5, or 6 ppb are 92800, 29136, 12832, 4256, 848, or 64 $km^2$, respectively in the original simulations at 4 km resolution. The area of the domain is 143856 $km^2$.

| Spatial resolution (km) | Reference HCHO volume mixing ratio (ppb) at 4 km resolution | | | | | |
|---|---|---|---|---|---|---|
| | 1 | 2 | 3 | 4 | 5 | 6 |
| 8 | 98 (%) | 95 | 91 | 84 | 79 | 0 |
| 12 | 97 | 92 | 85 | 73 | 79 | 0 |
| 20 | 97 | 86 | 72 | 67 | 58 | 0 |
| 36 | 97 | 82 | 52 | 29 | 0 | 0 |
| 48 | 96 | 72 | 51 | 0 | 0 | 0 |
| 100 | 96 | 62 | 0 | 0 | 0 | 0 |
| 200 | 89 | 56 | 0 | 0 | 0 | 0 |
| 300 | 53 | 46 | 0 | 0 | 0 | 0 |

*A number of details are missing about how the AMFs are computed beside the a priori profiles?*
*Angles, albedo, aerosols?*

→ The missing information is added in the revised manuscript. Solar zenith angles are 52.8°, 16.7°, and 28.8° at 16, 19, 22 UTC, respectively. Relative azimuth angles are 56.6°, 15.5°, 246.1° at 16, 19, 22 UTC, respectively. Viewing zenith angle in the VLIDORT is 46.5°. We assumed a constant surface reflectance of 0.05 across the domain. The AMF presented in the manuscript is selected at 340 nm similar to the current satellite retrieval. This information is included (Page 11, Line 9 – 19) in the revised manuscript.

*I think that the discussion about the shape factor introduce some confusion. I do not agree with the following sentence in the conclusion = "For similar profile shapes, the absolute magnitude of HCHO concentration is also an essential factor in determining the AMF". The author should clarify the impact of a change at a given altitude, that will modify the shape factor, in opposition to a change at all altitudes (multiplicative factor) that will not modify the shape factor and therefore have no impact on the AMF. See also the detailed comments. I would rather conclude that the AMF are very sensitive to the absolute HCHO mixing ratio in the boundary layer.*

→ We agree. Thank you for your comments. We changed the sentence to "Our study reveals that **the AMF is very sensitive to the absolute HCHO mixing ratio (or number density) in the boundary layer.** Therefore, the absolute magnitude of HCHO concentration in the boundary layer is an essential factor in determining the AMF".

I recommend publications after these comments have been addressed.

**Detailed comments**

*P2, l15: please quantify the statement "can better capture"*

→ We added a quantitative analysis in the sentence. Now it reads "…can better capture the spatial distributions of the enhanced HCHO plumes in an urban area than the nearly constant AMFs used for current operational products **by increasing the columns by ~50% in the domain-average and up to 100% at a finer scale**".

*P2, l16: This sentence is vague. Which operational product (reference?), what does "nearly constant AMF" mean?*

→ A reference on the SAO OMI HCHO product (Gonzalez Abad et al., 2015) is added. González Abad, G., Liu, X., Chance, K., Wang, H., Kurosu, T. P., and Suleiman, R. (2015), Updated Smithsonian Astrophysical Observatory Ozone Monitoring Instrument (SAO OMI) formaldehyde retrieval, *Atmos. Meas. Tech*., 8, 19-32, https://doi.org/10.5194/amt-8-19-2015.

*P3, l12: please cite Jin, X., Fiore, A. M., Murray, L. T., Valin, L. C., Lamsal, L. N., Duncan, B., Folkert Boersma, K., De Smedt, I., Abad, G. G., Chance, K. and Tonnesen, G. S.: Evaluating a space-based indicator of surface ozone-NO x -VOC sensitivity over mid-latitude source regions and application to decadal trends, J. Geophys. Res. Atmos., 439–461, doi:10.1002/2017JD026720, 2017.*

→ Jin et al. (2017) is added in the revised manuscript.

*p4, l6-10: HCHO weak absorption in the UV has an impact on slant column uncertainties. AMF uncertainties do not result from the weak HCHO absorption in the UV. Please clarify.*

→ Agreed. We modified structures of sentences to make the meaning clear. Now it reads as " Because of its weak absorption in the ultraviolet (UV) spectral region, HCHO is regarded as one of the most difficult species to retrieve from satellite-based radiance observations in the UV-visible (UV-VIS) spectral region (e.g., GOME/GOME-2, SCIAMACHY, OMI, and OMPS; see Martin et al., 2003, Zhu et al., 2016 for references). **In addition**, the **large** uncertainties in satellite trace gas retrievals based on UV-VIS spectral measurements arise from the calculation

of the air mass factor (AMF), which converts the slant column density of a trace gas to its vertical column values by considering the vertical sensitivity of the observations (AMF = slant column/vertical column, Palmer et al., 2001; Boersma et al., 2004; Lorente et al., 2017). **Therefore, it is important to identify factors affecting the accuracy of HCHO retrievals and to find a method to reduce these uncertainties**."

*P4, l18: add reference to operational products.*

→ Gonzalez Abad et al. (2015) is added in the revised manuscript.

*P4, l16: …, while the a priori profiles are generally derived from a 3D CTM.*

→ Corrected.

*P4, l18: which operational trace gas products? Please provide reference.*

→ References (Gonzalez Abad et al., 2015; De Smedt et al., 2017) are provided.

De Smedt, I., Theys, N., Yu, H., Danckaert, T., Lerot, C., Compernolle, S., Van Roozendael, M., Richter, A., Hilboll, A., Peters, E., Pedergnana, M., Loyola, D., Beirle, S., Wagner, T., Eskes, H., van Geffen, J., Boersma, K. F., and Veefkind, P.: Algorithm Theoretical Baseline for formaldehyde retrievals from S5P TROPOMI and from the QA4ECV project, Atmos. Meas. Tech. Discuss., https://doi.org/10.5194/amt-2017-393, in review, 2017.

*P5, l3-4: references are mixing satellite retrievals and inverse modelling papers.*

→ We modified the sentence. It now reads as "The HCHO retrievals from existing polar-orbiting satellites were investigated and **utilized** in previous studies…".

*P5, l5-6: It is not clear what is meant by this sentence "these studies …. used the contrast between land and ocean". Please add more explanations.*

→ This means that the detailed spatial variations in AMF in the US were not captured. We modified the sentence in the revised manuscript. Now it reads "these studies focused on regions with large biogenic sources or **showed large scale contrasts** between land and ocean."

*P5, l10: Please provide a reference for TROPOMI.*

→ Veefkind et al. (2012) is added in the revised manuscript.

*P5, l15: Recent model provide a resolution of 1x1°, daily (TM5-MP, TROPOMI)*

→ Now it is changed to "horizontal grid resolutions of **1**-3 degrees".

*P13, l1: Please specify to what quantity 35 % refers to. Total AMF, AMF in a certain altitude range?*

→ It meant a change in total AMF. We clarified it. It is changed to "Global Ozone Monitoring Experiment (GOME) measurements that were ~35% less sensitive to the HCHO column **(or 35% smaller total AMF)** over Tennessee than over the North Pacific."

*P13, l10: please provide a number (relative differences between cases a and b in figure A1) in order to estimate the "small" impact of surface pressure on AMF*

→ Quantitative analyses are shown in the revised manuscript. Please see our response to the other reviewer (Page 9–Page 11).

*P16, l15-16: I do not agree with this discussion. I completely agree that the AMF anti-correlates with the HCHO mixing ratio in the boundary layer. But if the absolute HCHO values changes in the boundary layer, and not at higher altitudes, this changes the profile shape quite strongly.*

→ Agreed. As suggested by the reviewer, we modified the sentences to "For UV-VIS retrievals, it is **well known** that the vertical profile shape affects the value of the AMF. Our study suggests a strong anti-correlation between the absolute concentration and the AMF: the AMF is low in the area of intense HCHO plumes. The changes in the absolute HCHO concentrations in the boundary layer (altitude AGL < 1-3 km) strongly modify profile shapes, which in turn affect AMF substantially."

*P23, l 8-9: quantify the improvement*

→ In the revised manuscript, we added quantitative analyses in several places. Therefore, we did not change this general conclusion.

*P 23, l8-8: It would be an interesting conclusion to provide a minimum resolution is time and in space, to reduce the AMF uncertainty under a given threshold (ex 10%).*

→ We suggested minimum resolution based on new analysis in Figure 7 and Table 1 (Figure R2 and Table R1 above). Because the simulations at 4 km resolution for the full domain of geostationary environmental satellite are very expensive, it is recommended to use 8-12 km if computing resources are available. More detailed discussions are added in the revised manuscript. See the responses above.

*Figure 1: Specify the dates in the legend*

→ We specified the dates (May-June 2010) in the legend in Figure 1.

*Figure 3: Please improve the visibility of the colorbar and the inset text.*

→ The visibility of the color bar and the text is improved in the revised manuscript.

*Figure 4: The altitude above ground level is not shown in this figure.*

→ The altitude above ground level (or AGL) is noted wherever needed.

*Figure 7: Please do not use "slope" factor. It introduces confusion. You already use profile shape and shape factor.*

→ "slope" factor is changed to "shape" factor in the manuscript and the figure. It is Figure 8 in the revised manuscript.

*After the paper from Palmer et al. 2001, several papers highlighted the importance of the a priori profile shapes on satellite HCHO retrieval: Barkley et al.,2012; De Smedt et al., 2015; Lorente et al., 2017; Wang et al., 2017.*

*Barkley, M. P., Kurosu, T. P., Chance, K. V, De Smedt, I., Van Roozendael, M., Arneth, A., Hagberg, D. and Guenther, A. B.: Assessing sources of uncertainty in formaldehyde air mass factors over tropical South America: Implications for top-down isoprene emission estimates, J. Geophys. Res., 117(D13), D13304, doi:10.1029/2011JD016827, 2012.*

[revised manuscript text omitted]